# Overcoming Missing Label Vocabulary in Black-Box Discrete Prompt Learning

## Abstract

Large language models (LLMs) have transformed natural language processing. While their scale challenges fine-tuning downstream tasks, prompt engineering offers a scalable, cost-effective solution to optimize their performance. Black-box prompt learning is crucial for leveraging the generative abilities of LLMs, especially in the Language-Model-as-a-Service scenario, where parameters and gradients are inaccessible. LLMs generate output exclusively in the form of encoded tokens processed through their backbone network. Existing black-box prompt learning methods rely on outputs corresponding to a predefined *label vocabulary*—a small subset of the token vocabulary of LLMs—to optimize prompts. However, in real-world applications, some datasets lack specific label vocabulary, and even manually assigned labels may perform inconsistently across different LLMs. To address these challenges, in this paper, we propose a novel label-vocabulary-free black-box discrete prompt learning method. Our approach employs an alternating optimization strategy to simultaneously learn discrete prompt tokens and a learnable matrix that directly maps the outputs of LLMs corresponding to the *token vocabulary* to categories. We provide theoretical convergence guarantees for our method under standard assumptions, ensuring its reliability. Experiments show that our method effectively learns prompts and outperforms existing baselines on datasets without label vocabulary.

## 1 Introduction

Large language models (LLMs) have revolutionized natural language processing (NLP) with their remarkable performance across various tasks, including text classification, machine translation, and dialogue (Touvron et al., 2023; Bubeck et al., 2023; Brown et al., 2020). For a given task, the user provides natural text input, which is tokenized according to a predefined token vocabulary for processing by a pre-trained LLM. The model then computes the most probable tokens from the vocabulary and decodes them back into human-readable text as output. A prompt, typically a sentence appended before or after a query input, can enhance the output quality of LLMs by guiding the model towards task-specific behavior without requiring additional training (Gao et al., 2021). This technique leverages the inherent knowledge embedded within pre-trained models to elicit desired responses and provides a cost-effective alternative to directly training or fine-tuning LLMs, making model adaptation both effective and efficient (Liu et al., 2023a; Chang et al., 2024).

Currently, companies developing LLMs typically offer only online application programming interfaces (APIs) for user interaction to safeguard their core technologies, a setup known as Language-Model-as-a-Service (LMaaS). In this context, users lack direct access to the model's parameters and gradients, resulting in an inevitable black-box scenario (Sun et al., 2022b). Within such a scenario, prompts become the only variables available for optimization (Diao et al., 2023). Consequently, optimizing prompts relies solely on probability evaluations from the LLM's API, necessitating the use of derivative-free methods.

Building upon these insights, several black-box prompt learning methodologies have emerged, demonstrating strong performance in text classification tasks. Continuous prompt learning approaches, such as BBT (Sun et al., 2022b), optimize continuous prompts that are prepended to the input text through derivative-free optimization within a low-dimensional embedding subspace. Furthermore, SSPT (Zhang et al., 2024) enhances this framework by employing subspace learning

and selection strategies to identify optimal low-dimensional subspaces within the BBT approach. However, continuous prompt learning exhibits limited applicability across diverse tasks. For instance, it cannot be directly applied to API prediction tasks that require discrete inputs. In contrast, discrete prompt learning methods, exemplified by BDPL (Diao et al., 2023), conceptualize prompt learning as a discrete token selection problem. In BDPL, prompt tokens are sampled from a categorical distribution and optimized using a policy gradient algorithm. Specifically, BDPL generates prompts based on their associated parameters, concatenates the tokenized prompt vectors with the tokenized sentence vectors, and feeds them into a LLM. The LLM's API then provides probability estimates for a predefined label vocabulary, which constitutes a small subset of the LLM's overall token vocabulary. These probability estimates are subsequently combined with one-hot label vectors to compute the objective function, which is then optimized using black-box optimization techniques.

While existing black-box discrete prompt learning are effective in scenarios with predefined label vocabularies, they face significant challenges when applied to real-world contexts where the label vocabulary is not predefined or may not align well with the LLM's token vocabulary. For instance, shopping websites generate data with rating preferences based on user-provided star ratings, such as those in the Amazon Books dataset (McAuley et al., 2015). These ratings are numerical values that do not directly correspond to the appropriate tokens within an LLM's vocabulary. As a result, it is not possible to directly obtain probability estimates for task categories from the LLM. Furthermore, when label words are missing, it is also cumbersome and difficult to use manual annotation to render existing black-box prompt learning methods effective across various downstream tasks. Therefore, a key problem remains underexplored: **how to optimize discrete prompts in black-box scenarios with missing label vocabulary.**

In this paper, we propose a novel label-vocabulary-free black-box discrete prompt learning method (LEAP) to address the problem. Specifically, we introduce a trainable matrix $M$ that serves as a learnable mapping mechanism, directly associating the LLM's output tokens with the desired task categories. This matrix effectively bridges the gap between the LLM's token vocabulary and the task-specific numerical value labels, allowing for flexible and adaptive prompt learning. Simultaneously, we employ an unbiased variance-reduced policy gradient approach to optimize the discrete prompt tokens. By leveraging policy gradient, we can iteratively refine the prompts based on the outputs from the LLM, ensuring that the prompts evolve in a direction that enhances task performance. A notable feature of our method is its end-to-end alternating optimization framework, which jointly learns the mapping matrix $M$ and the prompt parameters. This alternative optimization strategy ensures that both components evolve in harmony, leading to more coherent and effective prompt learning.

To the best of our knowledge, no previous studies have discussed how to learn prompts in the context of missing label vocabulary within the LMaaS scenario. We highlight the contributions and advantages of our work as follows:

- We introduce LEAP, a label-vocabulary-free black-box discrete prompt learning method that employs an innovative end-to-end alternating optimization framework. This framework jointly learns prompts and an output mapping matrix for LLMs, allowing both components to evolve in harmony and enhancing LLMs' adaptability in scenarios where label vocabulary is missing.
- We provide a rigorous convergence analysis of our optimization framework, demonstrating that LEAP achieves a convergence complexity $\mathcal{O}\left(\frac{1}{\epsilon^4}\right)$ under standard assumptions. Our theoretical analysis highlights that the variance occurring during the alternating process is controlled by the prompt's sampling times and mini-batch size, thereby guaranteeing the efficacy of our approach in label-free prompt learning.
- We conduct an extensive evaluation of our approach across multiple LLMs to ensure its generalizability. The experimental results show that our method outperforms baseline methods, highlighting its effectiveness in scenarios where label vocabulary is missing.

## 2 RELATED WORK

### 2.1 PROMPT LEARNING

Prompt learning has recently gained prominence as a powerful paradigm in natural language processing, leveraging pre-trained language models to tackle a wide range of downstream tasks with

minimal task-specific training data. Early work in this field centered on prompt engineering, where manually crafted prompts were employed to guide language models toward producing desired behaviors (Petroni et al., 2019; Schick & Schütze, 2021). These handcrafted prompts, while effective, often necessitated considerable expertise and domain knowledge. To mitigate this limitation, researchers developed prompt tuning techniques that automate the optimization of prompts by learning optimal representations. Notable works in this area, such as P-tuning (Liu et al., 2023b), Prefix-tuning (Li & Liang, 2021), P-tuning V2 (Liu et al., 2021), and Prompt-tuning (Lester et al., 2021), focus on learning continuous embeddings of soft prompts with tunable parameters.

### 2.2 Black-Box Prompt Learning

Despite the success of prompt tuning in white-box settings, where model parameters and gradients are accessible, there has been increasing interest in black-box prompt learning. This approach is particularly pertinent in scenarios where language models are offered as services via APIs, restricting user access to the model's internal mechanisms. In these black-box environments, the primary challenge lies in optimizing prompts based solely on the model's outputs, without the capability to directly modify or fine-tune the model's parameters.

Several significant works have been proposed to tackle this challenge, which can be primarily categorized into two paradigms: continuous prompt learning and discrete prompt learning. BBT (Sun et al., 2022b) and BBTv2 (Sun et al., 2022a) utilize Covariance Matrix Adaptation Evolution Strategy (CMA-ES) to optimize continuous prompt embeddings within a low-dimensional embedding subspace. SSPT (Zhang et al., 2024) incorporates subspace learning and selection techniques to identify the optimal low-dimensional subspace within BBT. However, the learned continuous prompt embeddings are less interpretable compared to discrete prompts and cannot be directly applied to prediction APIs that only accept discrete inputs. This limitation significantly restricts their practical usability in many real-world applications.

In contrast, black-box discrete prompt tuning emphasizes the optimization of human-readable and interpretable prompts, which are directly applicable in scenarios where only discrete text inputs are accepted, such as prediction APIs. Discrete prompts offer the dual advantages of interpretability and deployability in real-world applications without necessitating additional processing steps. Building on this foundation, RLPrompt (Deng et al., 2022) employs reinforcement learning to optimize discrete prompts in black-box settings. By framing the prompt optimization process as a reinforcement learning problem, RLPrompt iteratively refines prompts based on feedback derived from the model's outputs. GAP3 (Zhao et al., 2023) is a genetic algorithm that evolves discrete prompts from empty templates by leveraging predictive probabilities from large pre-trained language models, thereby eliminating the need for manual prompts or API injections. Additionally, BDPL (Diao et al., 2023) utilizes the policy gradient method to optimize the categorical distribution of prompt vocabularies.

## 3 Methodology

In this section, we first introduce the proposed alternating optimization framework from an overall perspective and explain how it facilitates black-box discrete prompt learning without relying on a label vocabulary. Next, the unbiased variance-reduced policy gradient descent for optimizing discrete prompt tokens and the proximal gradient descent for optimizing the mapping matrix $M$ are given in detail, respectively. Finally, we provide a detailed description of the algorithmic pipeline.

### 3.1 Overall Framework

**Notations.** Let $\widetilde{\mathcal{D}} \triangleq \{(\boldsymbol{s}_1, y_1), (\boldsymbol{s}_2, y_2), \ldots, (\boldsymbol{s}_K, y_K)\}$ denote the training dataset with cardinality $K$. For each $k \in \{1, 2, \ldots, K\}$, $\boldsymbol{s}_k$ represents an input training example (e.g., a piece of text), and $y_k \in \{1, \ldots, C\}$ denotes its corresponding label, where $C$ is the number of categories. We define $\text{Tok}(\cdot)$ as a tokenizer that converts input text into a token vector, and let $\boldsymbol{x}_k \triangleq \text{Tok}(\boldsymbol{s}_k)$ denote the $k$-th token vector. The label $y_k$ is represented as a one-hot encoded vector $\boldsymbol{y}_k$. Let $\mathcal{D} \triangleq \{\boldsymbol{d}_1, \boldsymbol{d}_2, \ldots, \boldsymbol{d}_K\}$ denote the set of tuples composed of token vectors and their corresponding one-hot labels, where $\boldsymbol{d}_k = (\boldsymbol{x}_k, \boldsymbol{y}_k)$ represents an individual sample. $\boldsymbol{M} = (m_{d,c})_{D \times C}$ signifies the mapping matrix.

**Black-box Discrete Prompt Learning.** Discrete black-box prompt learning aims to learn a discrete textual prompt consisting of $n$ tokens, denoted by $\Phi = \phi_1...\phi_i...\phi_n = \mathcal{V}[j_1]...\mathcal{V}[j_i]...\mathcal{V}[j_n]$, where $\mathcal{V} = (\mathcal{V}[j])_{j=1}^N$ represents the vocabulary list for the prompt, and $\phi_i = \mathcal{V}[j_i]$ is the $i$-th token in $\Phi$, corresponding to the $j$-th token in $\mathcal{V}$. We assume that each prompt index $j_i$ follows an independent categorical distribution, i.e., $j_i \sim \text{Cat}(\boldsymbol{p}_i)$, where the random variable $j_i$ is sampled according to the probability distribution $\boldsymbol{p}_i = [p_{i,1}, p_{i,2}, ..., p_{i,N}]$ over the $N$ token indices. Here, $\boldsymbol{p}_i \in \mathcal{C}$ and $\mathcal{C} = \{\boldsymbol{p} : \|\boldsymbol{p}\|_1 = 1, 0 \preceq \boldsymbol{p} \preceq 1\}$. Given the independence of each $\boldsymbol{p}_i$, the joint probability of the entire discrete prompt is given by $\mathcal{P}(\Phi) = \prod_{i=1}^n p_{i,j_i}$.

**Missing Label Vocabulary Problem.** Although black-box discrete prompt learning can effectively optimize prompt without requiring an in-depth understanding of the internal mechanisms of LLMs, existing black-box prompt learning approaches rely on outputs aligned with a predefined label vocabulary to optimize prompts. However, in practical applications, certain datasets may lack specific label vocabulary, and even manually assigned labels can demonstrate inconsistent performance across various LLMs. Therefore, our objective is to perform discrete, label-free prompt optimization within black-box scenarios. Specifically, we employ a mapping matrix $\boldsymbol{M}$ that directly maps the outputs of LLMs corresponding to their token vocabulary to predefined categories. Additionally, incorporating $\ell_1$-regularization into the mapping matrix enhances sparsity, thereby enabling more efficient selection of the most relevant features within $\boldsymbol{M}$. We define the loss function: $\mathcal{L}(\Phi, \boldsymbol{M}; \mathcal{D}) \triangleq \mathcal{L}(\text{Softmax}(\mathcal{G}(\Phi, \boldsymbol{X})) \cdot \boldsymbol{M}, \boldsymbol{Y})$. The objective function can be expressed as:

$$\min_{\Phi, \boldsymbol{M}} F(\Phi, \boldsymbol{M}; \mathcal{D}) \triangleq \mathbb{E}_\Phi [\mathcal{L}(\Phi, \boldsymbol{M}; \mathcal{D})] + r(\boldsymbol{M}). \tag{1}$$

where $\mathcal{G}$ represents the LLM model, $\mathcal{L}$ denotes the loss function, and $r(\boldsymbol{M}) = \lambda \cdot \|\boldsymbol{M}\|_1$ denotes the $\ell_1$-regularization applied to $\boldsymbol{M}$.

**Alternating Optimization.** We propose a Label-vocabulary-free Black-box Discrete Prompt Learning (LEAP), an end-to-end alternating optimization framework specifically designed for prompt learning. Initially, we conceptualize the prompt learning process as a discrete token selection problem, where appropriate prompt tokens are sampled based on the classification distribution. This approach allows for the optimization of prompt tokens independently from the parameters and gradients of the pre-trained model. To enhance stability, we employ a unbiased variance-reduced policy gradient estimator to optimize the categorical distribution of prompt $\Phi$, thereby mitigating the instability caused by the high variance inherent in prompt sampling. Subsequently, we optimize the mapping matrix $\boldsymbol{M}$ by incorporating $\ell_1$-regularization terms that promote feature sparsification and reduce redundant information. Our alternating optimization framework alleviates the complexity associated with jointly optimizing $\Phi$ and $\boldsymbol{M}$, enabling the focused optimization of each parameter individually and thereby improving the overall performance of the model.

### 3.2 Unbiased variance-reduced Policy Gradient Descent

**Gumbel-Softmax reparameterization.** We re-parameterize the categorical distribution $\text{Cat}(\boldsymbol{P})$ of the prompt with the Gumbel-Softmax ($\mathcal{S}$) function (Jang et al., 2016):

$$p_{i,j} = \mathcal{S}(\alpha_{i,j}) = \frac{\exp\left(\frac{\log(\alpha_{i,j}) + g_{i,j}}{\tau}\right)}{\sum_{\rho=1}^N \exp\left(\frac{\log(\alpha_{i,\rho}) + g_{i,\rho}}{\tau}\right)}, \tag{2}$$

where $\boldsymbol{P} = (p_{i,j})_{n \times N} \in \mathbb{R}^{n \times N}$ is the sampling probability matrix for $\Phi$, $\alpha_{i,j} > 0$ is learnable parameters and $\boldsymbol{\alpha} \in \mathbb{R}^{n \times N}$, $\tau > 0$ is the temperature parameter, $g_{i,j}$ is sampled from the Gumbel$(0, 1)$. The reparameterization of the categorical distribution uses the Gumbel-Softmax technique to mitigate bias (**Lemma 3**) that is typically associated with the direct optimization of probability distributions in (Diao et al., 2023).

**Policy Gradient Estimator.** Leveraging Gumbel-Softmax reparameterization and policy gradient estimator, to optimize loss with the forward propagation, $\mathbb{E}_\Phi [\mathcal{L}(\Phi, \boldsymbol{M}; \mathcal{D})]$ can be expressed as:

$$\mathbb{E}_{\Phi \sim \mathcal{S}(\boldsymbol{\alpha})} [\mathcal{L}(\Phi, \boldsymbol{M}; \mathcal{D})] = \sum_{\phi_1 \sim \mathcal{S}(\boldsymbol{\alpha}_1)} \cdots \sum_{\phi_n \sim \mathcal{S}(\boldsymbol{\alpha}_n)} \left[ \mathcal{L}(\Phi, \boldsymbol{M}; \mathcal{D}) \cdot \prod_{i=1}^n \mathcal{P}(\phi_i) \right]. \tag{3}$$

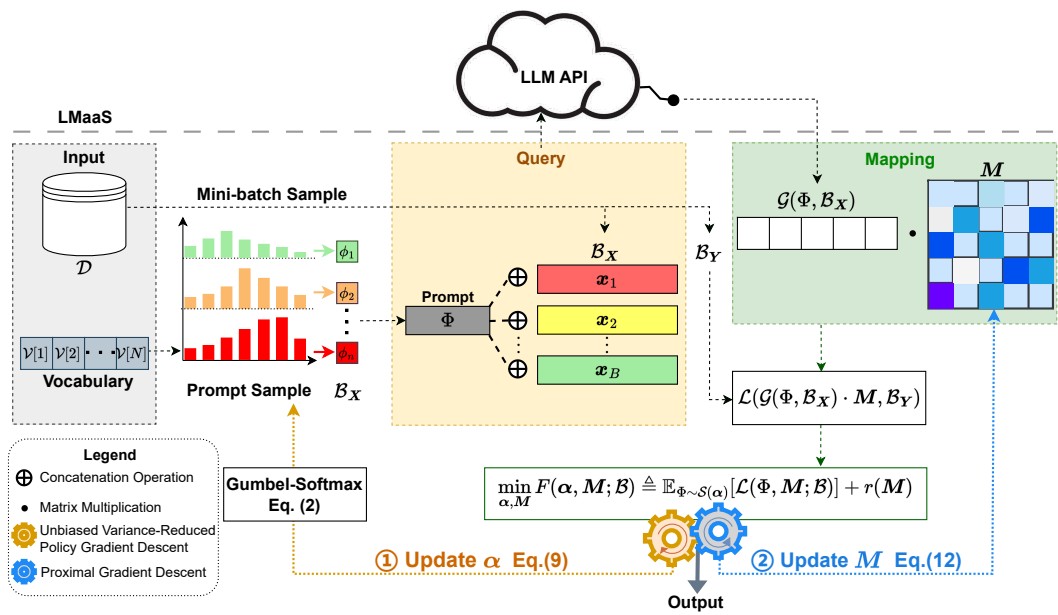

Figure 1: Our proposed framework for label-vocabulary-free black-box discrete prompt learning. We first concatenate the prompts $\Phi$ with each input token $x$ in the mini-batch to create the query input for the LLM. The prompts are sampled from the prompt vocabulary according to a categorical distribution, with the probabilities of this distribution derived from the Gumbel-Softmax operation applied to the parameter $\boldsymbol{\alpha}$. Subsequently, we obtain the probabilities output from the LLM API. Finally, we utilize a sparsified matrix $M$ to directly map the LLM's outputs to the corresponding categories. **Update Mechanism:** Update $\boldsymbol{\alpha}$ (yellow gear)-unbiased variance-reduced gradient descent is employed to update $\boldsymbol{\alpha}$, using the mapping matrix $M$ from the recent update. Update $M$ (blue gear)-proximal gradient descent is applied to update $M$, where the prompts are sampled based on the categorical distributions generated from the current update of $\boldsymbol{\alpha}$.

Since the optimization variable is $\boldsymbol{\alpha}$, we can redefine the objective function (1) as follows:

$$\min_{\boldsymbol{\alpha},\boldsymbol{M}} F(\boldsymbol{\alpha},\boldsymbol{M};\mathcal{D}) \triangleq \mathbb{E}_{\Phi\sim\mathcal{S}(\boldsymbol{\alpha})}\left[\mathcal{L}(\Phi,\boldsymbol{M};\mathcal{D})\right] + r(\boldsymbol{M}). \tag{4}$$

Then, we can estimate the gradient of $\boldsymbol{\alpha}_i$ as follows:

$$\nabla_{\boldsymbol{\alpha}_i} F(\boldsymbol{\alpha},\boldsymbol{M};\mathcal{D}) = \nabla_{\boldsymbol{\alpha}_i}\mathbb{E}_{\Phi\sim\mathcal{S}(\boldsymbol{\alpha})}[\mathcal{L}(\Phi,\boldsymbol{M};\mathcal{D})]$$

$$= \sum_{\phi_1\sim\mathcal{S}(\boldsymbol{\alpha}_1)}\cdots\sum_{\phi_n\sim\mathcal{S}(\boldsymbol{\alpha}_n)}\left[\mathcal{L}(\Phi,\boldsymbol{M};\mathcal{D})\cdot\nabla_{\boldsymbol{\alpha}_i}\prod_{i=1}^{n}\mathcal{P}(\phi_i)\right]$$

$$= \sum_{\phi_1\sim\mathcal{S}(\boldsymbol{\alpha}_1)}\cdots\sum_{\phi_n\sim\mathcal{S}(\boldsymbol{\alpha}_n)}[\mathcal{L}(\Phi,\boldsymbol{M};\mathcal{D})\cdot\nabla_{\boldsymbol{\alpha}_i}\mathcal{P}(\phi_i)]$$

$$= \sum_{\phi_1\sim\mathcal{S}(\boldsymbol{\alpha}_1)}\cdots\sum_{\phi_n\sim\mathcal{S}(\boldsymbol{\alpha}_n)}[\mathcal{L}(\Phi,\boldsymbol{M};\mathcal{D})\cdot\nabla_{\boldsymbol{\alpha}_i}\log(\mathcal{P}(\phi_i))\cdot\mathcal{P}(\phi_i)]$$

$$= \mathbb{E}_{\Phi\sim\mathcal{S}(\boldsymbol{\alpha})}[\mathcal{L}(\Phi,\boldsymbol{M};\mathcal{D})\cdot\nabla_{\boldsymbol{\alpha}_i}\log\mathcal{P}(\phi_i)]. \tag{5}$$

Considering $\phi_i = \mathcal{V}[j_i]$, we can give explicitly $\nabla_{\boldsymbol{\alpha}_i}\log\mathcal{P}(\phi_i)$ as follow:

$$\nabla_{\alpha_{i,j}}\log\mathcal{P}(\phi_i) = \nabla_{\alpha_{i,j}}\log p_{i,j_i} = \begin{cases} \frac{1-p_{i,j_i}}{\tau\alpha_{i,j_i}}, & j = j_i \\ -\frac{p_{i,j}}{\tau\alpha_{i,j}}, & j \neq j_i \end{cases}. \tag{6}$$

**Unbiased Mini-batch Stochastic Variance-Reduced Policy Gradient Estimator.** Let $\mathcal{B}$ be the mini-batch sampled from $\Psi$ and $B$ is the batch size, then the mini-batch stochastic variance-reduced

policy gradient is computed:

$$\mathcal{L}_{avg} = \frac{1}{I_{\boldsymbol{\alpha}}} \sum_{r=1}^{I_{\boldsymbol{\alpha}}} \mathcal{L}(\Phi^r, \boldsymbol{M}; \mathcal{B}), \tag{7}$$

$$\hat{\nabla}_{\boldsymbol{\alpha}_i} f_{\mathcal{B}}(\boldsymbol{\alpha}, \boldsymbol{M}) = \frac{1}{I_{\boldsymbol{\alpha}} - 1} \sum_{r=1}^{I_{\boldsymbol{\alpha}}} \left( \mathcal{L}(\Phi^r, \boldsymbol{M}, \mathcal{B}) - \mathcal{L}_{avg} \right) \cdot \nabla_{\boldsymbol{\alpha}_i} \log \mathcal{P}(\phi_i), \tag{8}$$

where $\{\Phi^r\}_{r=1}^{I_{\boldsymbol{\alpha}}}$ are sampled independently from $\mathcal{V}$ through categorical distribution $\mathrm{Cat}(\mathcal{S}(\boldsymbol{\alpha}))$. Consequently, when the learning rate is set to $\eta_{\boldsymbol{\alpha}}$, the update of $\boldsymbol{\alpha}_i$ can be formulated as follows:

$$\boldsymbol{\alpha}_{i,t+1} = \boldsymbol{\alpha}_{i,t} - \eta_{\boldsymbol{\alpha}} \cdot \hat{\nabla}_{\boldsymbol{\alpha}_i} f_{\mathcal{B}}(\boldsymbol{\alpha}_t, \boldsymbol{M}_t), i = 1, ..., n. \tag{9}$$

### 3.3 PROXIMAL GRADIENT DESCENT FOR THE MAPPING MATRIX

First, we independently sample $\{\Phi^s\}_{s=1}^{I_M}$ from $\mathcal{V}$ using the categorical distribution $\mathrm{Cat}(\mathcal{S}(\boldsymbol{\alpha}))$, and compute the gradient of $\mathbb{E}_{\Phi \sim \mathcal{S}(\boldsymbol{\alpha})} [\mathcal{L}(\Phi, \boldsymbol{M}; \mathcal{D})]$ with respect to $\boldsymbol{M}$ as follows:

$$\tilde{\nabla}_{\boldsymbol{M}} f_{\mathcal{B}}(\boldsymbol{\alpha}, \boldsymbol{M}) = \nabla_{\boldsymbol{M}} \left( \frac{1}{I_M} \sum_{s=1}^{I_M} \mathcal{L}(\Phi^s, \boldsymbol{M}; \mathcal{B}) \right). \tag{10}$$

We subsequently apply $\ell_1$-regularization to induce sparsity in $\boldsymbol{M}$. Specifically, we note $r(\boldsymbol{M})$ is convex and sufficiently simple to ensure the existence of its proximal mapping:

$$\mathrm{prox}_{\eta_M r}[\boldsymbol{M}] = \arg\min_{\boldsymbol{A}} \left\{ \frac{1}{2\eta_M} \|\boldsymbol{A} - \boldsymbol{M}\|^2 + r(\boldsymbol{A}) \right\}. \tag{11}$$

Consequently, when the learning rate is set to $\eta_M$, for each iteration $t = 0, ..., T - 1$, we employ proximal gradient descent to update $\boldsymbol{M}$:

$$\boldsymbol{M}_{t+1} \in \mathrm{prox}_{\eta_M r} \left[ \boldsymbol{M}_t - \eta_M \cdot \tilde{\nabla}_{\boldsymbol{M}} f_{\mathcal{B}}(\boldsymbol{\alpha}_{t+1}, \boldsymbol{M}_t) \right]. \tag{12}$$

### 3.4 ALGORITHMIC PIPELINE OF LEAP

By alternately updating $\boldsymbol{\alpha}$ and $\boldsymbol{M}$, the proposed algorithm is presented in **Algorithm 1** and a single update round is illustrated in **Figure 1**. The training process for each iteration is as follows. First, a mini-batch $\mathcal{B}$ and a set of prompts $\{\Phi^r\}_{r=1}^{I_{\boldsymbol{\alpha}}}$ are obtained by sampling from $\mathcal{D}$ and $\mathrm{Cat}(\mathcal{S}(\boldsymbol{\alpha}))$, respectively. The corresponding losses are then computed, and $\boldsymbol{\alpha}$ is updated using unbiased variance-reduced policy gradient descent. Next, the updated $\boldsymbol{\alpha}$ is employed to generate a new set of prompt samples $\{\Phi^s\}_{s=1}^{I_M}$, after which $\boldsymbol{M}$ is updated via proximal gradient descent. This process completes the updates of both $\boldsymbol{\alpha}$ and $\boldsymbol{M}$.

## 4 CONVERGENCE ANALYSIS

### 4.1 ASSUMPTION

**Assumption 1** (Bounded variance of stochastic gradients). *The stochastic gradients is unbiased and we assume the variance of stochastic gradients for $\boldsymbol{\alpha}_i$ and $\boldsymbol{M}$ is bounded:*

$$\mathbb{E}_{(\boldsymbol{x}_k, \boldsymbol{y}_k) \in \mathcal{D}} \left\| \nabla_{\boldsymbol{\alpha}_i} f_k(\boldsymbol{\alpha}, \boldsymbol{M}) - \mathbb{E}_{(\boldsymbol{x}_k, \boldsymbol{y}_k) \in \mathcal{D}} \left[ \nabla_{\boldsymbol{\alpha}_i} f_k(\boldsymbol{\alpha}, \boldsymbol{M}) \right] \right\|_2^2 \le \sigma_{\boldsymbol{\alpha}}^2; \tag{13}$$

$$\mathbb{E}_{(\boldsymbol{x}_k, \boldsymbol{y}_k) \in \mathcal{D}} \left\| \nabla_{\boldsymbol{M}} f_k(\boldsymbol{\alpha}, \boldsymbol{M}) - \mathbb{E}_{(\boldsymbol{x}_k, \boldsymbol{y}_k) \in \mathcal{D}} \left[ \nabla_{\boldsymbol{M}} f_k(\boldsymbol{\alpha}, \boldsymbol{M}) \right] \right\|_2^2 \le \sigma_{\boldsymbol{M}}^2. \tag{14}$$

**Assumption 2** (Lower Boundedness for objective function). *Given an initial point $(\boldsymbol{\alpha}_0, \boldsymbol{M}_0)$, $(\boldsymbol{\alpha}_*, \boldsymbol{M}_*)$ denotes the global minimum of $F(\boldsymbol{\alpha}, \boldsymbol{M}; \mathcal{D})$, there exists $\triangle < \infty$ such that*

$$F(\boldsymbol{\alpha}_0, \boldsymbol{M}_0; \mathcal{D}) - F(\boldsymbol{\alpha}_*, \boldsymbol{M}_*; \mathcal{D}) \le \triangle. \tag{15}$$

**Assumption 3** (Bounded Loss). *We clip loss function with a constant $G$:*

$$|\mathcal{L}(\Phi, \boldsymbol{M}; \mathcal{D})| \le U. \tag{16}$$

**Algorithm 1:** Label-vocabulary-free Black-box Discrete Prompt Learning (LEAP)

---

**Input:** Training dataset $\mathcal{D}$;
         Learning rates $\eta_{\boldsymbol{\alpha}}$ and $\eta_M$;
         Sampling times $I_{\boldsymbol{\alpha}}$ and $I_M$.
**Output:** The learned parameters $\boldsymbol{\alpha}_T$ and $\boldsymbol{M}_T$.

1   Initial parameters $\boldsymbol{\alpha}_0$ and $\boldsymbol{M}_0$
2   **for** $t = 0, 1, \ldots, T - 1$ **do**
3      |   $\mathcal{B}_t \leftarrow$ split $\mathcal{D}$ into mini-batch of size $B$
      |   `// Update `$\boldsymbol{\alpha}$
4      |   **for** $r = 1, 2, \ldots, I_{\boldsymbol{\alpha}}$ **do**
5      |      | Get $\{\mathcal{L}(\Phi_t^r, \boldsymbol{M}_t; \mathcal{B}_t)\}_{r=1}^{I_{\boldsymbol{\alpha}}}$ by sampling $\{\Phi_t^r\}_{r=1}^{I_{\boldsymbol{\alpha}}}$ from $\mathcal{V}$ through $\mathrm{Cat}(\mathcal{S}(\boldsymbol{\alpha}_t))$
6      |   **for** $i = 1, 2, \ldots, n$ **do**
7      |      | $\mathcal{L}_{avg} = \frac{1}{I_{\boldsymbol{\alpha}}} \sum_{r=1}^{I_{\boldsymbol{\alpha}}} \mathcal{L}(\Phi_t^r, \boldsymbol{M}_t; \mathcal{B}_t)$
8      |      | $\hat{\nabla}_{\boldsymbol{\alpha}_i} f_{\mathcal{B}}(\boldsymbol{\alpha}_t, \boldsymbol{M}_t) = \frac{1}{I_{\boldsymbol{\alpha}}-1} \sum_{r=1}^{I_{\boldsymbol{\alpha}}} (\mathcal{L}(\Phi_t^r, \boldsymbol{M}_t; \mathcal{B}_t) - \mathcal{L}_{avg}) \cdot \nabla_{\boldsymbol{\alpha}_i} \log \mathcal{S}(\boldsymbol{\alpha}_t)$
9      |      | $\boldsymbol{\alpha}_{i,t+1} = \boldsymbol{\alpha}_{i,t+1} - \eta_{\boldsymbol{\alpha}} \cdot \hat{\nabla}_{\boldsymbol{\alpha}_i} f_{\mathcal{B}}(\boldsymbol{\alpha}_t, \boldsymbol{M}_t)$
      |   `// Update `$\boldsymbol{M}$
10     |   **for** $s = 1, 2, \ldots, I_M$ **do**
11     |      | Get $\left\{\mathcal{L}(\Phi_{t+1}^s, \boldsymbol{M}_t; \mathcal{B}_t)\right\}_{s=1}^{I_M}$ by sampling $\left\{\Phi_{t+1}^s\right\}_{s=1}^{I_M}$ from $\mathcal{V}$ through $\mathrm{Cat}(\mathcal{S}(\boldsymbol{\alpha}_{t+1}))$
12     |   $\tilde{\nabla}_{\boldsymbol{M}} f_{\mathcal{B}}(\boldsymbol{\alpha}_{t+1}, \boldsymbol{M}_t) = \nabla_{\boldsymbol{M}}(\frac{1}{I_M} \sum_{s=1}^{I_M} \mathcal{L}(\Phi_{t+1}^s, \boldsymbol{M}_t, \mathcal{B}_t))$
13     |   $\boldsymbol{M}_{t+1} \in \mathrm{prox}_{\eta_M r}\left[\boldsymbol{M}_t - \eta_M \cdot \tilde{\nabla}_{\boldsymbol{M}} f_{\mathcal{B}}(\boldsymbol{\alpha}_{t+1}, \boldsymbol{M}_t)\right]$

---

**Assumptions 1 and 2** constitute the foundational premises for addressing non-convex optimization problems using stochastic gradient descent, as demonstrated in prior studies (Ghadimi & Lan, 2013; Hazan & Kale, 2014; Xu et al., 2019; Liu et al., 2020). **Assumption 3** ensures that the loss function remains bounded by regulating the loss during the estimation of the $I_{\boldsymbol{\alpha}}$-th and $I_M$-th samples when updating $\boldsymbol{\alpha}$ and $\boldsymbol{M}$. This boundedness is essential for facilitating rigorous theoretical analysis. It is important to recognize that loss functions, such as the cross-entropy function, can potentially become unbounded. In practical applications, these loss values are typically clipped to maintain boundedness.

### 4.2 CONVERGENCE ANALYSIS OF LEAP

**Theorem 1** (Convergence of LEAP). *Suppose **Assumption 1**, **2** and **3** hold, for iteration $t = 0, ..., T - 1$, set $\alpha_{i,j} \geq \beta > 0$ and $|m_{d,c}| \geq \xi > 0$, $\tau > 0$ is the temperature parameter, $f_{\mathcal{D}}(\boldsymbol{\alpha}, \boldsymbol{M})$ is smooth for $\boldsymbol{\alpha}$ with smooth constant $L_{\boldsymbol{\alpha}} = \frac{nUN(\tau+1)}{\tau^2 \beta^2}$ and lipschitz smooth for $\boldsymbol{M}$ with smooth constant is $L_M = \frac{1}{\xi^2}$, $\sigma_{\boldsymbol{\alpha}}^2$ and $\sigma_M^2$ are the variance of the stochastic gradient for $\boldsymbol{\alpha}$ and $\boldsymbol{M}$, $\tilde{\sigma}_{\boldsymbol{\alpha}}^2 = \frac{8U^2 N}{\tau^2 \beta^2}$ and $\tilde{\sigma}_M^2 = \frac{4}{\xi^2}$ are the variance of prompt sampling for $\boldsymbol{\alpha}$ and $\boldsymbol{M}$. We define $\eta_{min} = \min\{\eta_{\boldsymbol{\alpha}}, \eta_M\}$ and $\eta_{max} = \max\{\eta_{\boldsymbol{\alpha}}, \eta_M\}$, and run **Algorithm 1** with $0 < \eta_{\boldsymbol{\alpha}} < \frac{1}{L_{\boldsymbol{\alpha}}}$, $0 < \eta_M < \frac{1}{L_M}$ and $q_{\eta} = \frac{\eta_{max}}{\eta_{min}} < \infty$, then the following inequality holds:*

$$\frac{1}{T} \sum_{t=0}^{T-1} \left( \|\nabla_{\boldsymbol{\alpha}} f_{\mathcal{D}}(\boldsymbol{\alpha}_t, \boldsymbol{M}_t)\|_2^2 + \|g_{\mathcal{D}}(\boldsymbol{\alpha}_{t+1}, \boldsymbol{M}_t)\|_2^2 \right)$$

$$\leq \frac{2\triangle}{T\eta_{min}} + \frac{2nq_{\eta}\tilde{\sigma}_{\boldsymbol{\alpha}}^2}{I_{\boldsymbol{\alpha}}^2} + \frac{4q_{\eta}\tilde{\sigma}_M^2}{I_M} + \frac{2nq_{\eta}\sigma_{\boldsymbol{\alpha}}^2 + 4q_{\eta}\sigma_M^2}{B}. \tag{17}$$

*where $\nabla_{\boldsymbol{\alpha}} f_{\mathcal{D}}(\boldsymbol{\alpha}_t, \boldsymbol{M}_t)$ is the full gradient for $\boldsymbol{\alpha}$, and $g_{\mathcal{D}}(\boldsymbol{\alpha}_{t+1}, \boldsymbol{M}_t)$ is the gradient mapping of full gradient for $\boldsymbol{M}$ (24).*

**Remark 1**. We clip the lower bounds for $\alpha_{i,j}$ and $m_{d,c}$ respectively: 1) The clipping for $\alpha_{i,j}$ is because the Gumbel-Softmax function has the term $\log(\alpha_{i,j})$ (2), which naturally requires $\alpha_{i,j} > 0$, which is necessary for both the experimental setup and theoretical analysis. 2) The clipping for $m_{d,c}$

is essentially guaranteeing that the lower bound of $\sum_{d=1}^{D} [\mathcal{G}_{k,d} \cdot m_{d,c^*}]$ is not 0 in (32) of **Lemma 2** and (40) of **Lemma 4**, when $\sum_{d=1}^{D} (\mathcal{G}_{k,d} \cdot m_{d,c^*}) = 0$, the cross-entropy loss function (30)(31) $\mathcal{L}(\Phi, \boldsymbol{M}; \boldsymbol{d}_k) = -\boldsymbol{y}_k \cdot [\log(\mathcal{G}_k \cdot \boldsymbol{M})]^\top = -\log\left[\sum_{d=1}^{D} (\mathcal{G}_{k,d} \cdot m_{d,c^*})\right]$ appears to be infinity, and hence we bound the lower bound of $|m_{d,c}|$.

**Corollary 1** (Convergence complexity of LEAP). *Suppose **Assumption 1, 2** and **3** hold, and run **Algorithm 1** with $\eta_{\boldsymbol{\alpha}} = \frac{c_1}{L_{\boldsymbol{\alpha}}}(0 < c_1 < 1)$, $\eta_{\boldsymbol{M}} = \frac{c_2}{L_M}(0 < c_1 < 1)$, $\eta_{min} = \min\left\{\frac{c_1}{L_{\boldsymbol{\alpha}}}, \frac{c_2}{L_M}\right\}, q_\eta = \max\left\{\frac{c_1}{c_2}, \frac{c_2}{c_1}\right\} < \infty$, $B = \frac{8nq_\eta\sigma_{\boldsymbol{\alpha}}^2 + 16q_\eta\sigma_M^2}{\epsilon^2}$, $I_{\boldsymbol{\alpha}} = \frac{\sqrt{8nq_\eta\tilde{\sigma}_{\boldsymbol{\alpha}}^2}}{\epsilon}$, $I_{\boldsymbol{M}} = \frac{16q_\eta\tilde{\sigma}_M^2}{\epsilon^2}$ and $T = \frac{8\triangle}{\eta_{min}\epsilon^2}$, then the output of **Algorithm 1** satisfies:*

$$\frac{1}{T}\sum_{t=0}^{T-1}\left(\|\nabla_{\boldsymbol{\alpha}} f_{\mathcal{D}}(\boldsymbol{\alpha}_t, \boldsymbol{M}_t)\|_2^2 + \|g_{\mathcal{D}}(\boldsymbol{\alpha}_{t+1}, \boldsymbol{M}_t)\|_2^2\right) \le \epsilon^2. \tag{18}$$

*Thus, the total oracle complexity for LEAP is $\mathcal{O}\left(\frac{1}{\epsilon^4}\right)$.*

**Proof skeleton:** For the LEAP algorithm: We begin by establishing the Lipschitz smoothness (**Lemma 1** and **2**) of the objective function with respect to the parameters $\boldsymbol{\alpha}$ and $\boldsymbol{M}$, based on the clipped loss function (**Assumption 3**). This Lipschitz smoothness is a crucial prerequisite for analyzing the nonconvex optimization problem. Subsequently, we examine two sources of stochasticity in the alternating optimization process: the stochasticity introduced by mini-batch sampling (**Assumption 1**) and the randomness inherent in prompt sampling (**Lemmas 3** and **4**). Building on these foundational assumptions and lemmas, we then prove the convergence of the LEAP algorithm (**Theorem 1**) and analyze its convergence complexity (**Corollary 1**). The proof of **Lemmas 1-4** are in the Appendix A.4. The proof of **Theorem 1** and **Corollary 1** are in the Appendix A.5.

## 5 EXPERIMENTS

### 5.1 EXPERIMENT SETUPS

**Datasets.** To evaluate the performance of our method, we conduct experiments using eight datasets: BOOK (McAuley et al., 2015), CoLA (Warstadt et al., 2019), ELEC (McAuley et al., 2015), QNLI (Wang et al., 2019), RTE (Dagan et al., 2005), SNLI (Bowman et al., 2015), SST-2 (Socher et al., 2013), and AG (Zhang et al., 2015). These datasets cover a variety of standard language understanding tasks. Detailed descriptions of these datasets are given in the Appendix A.6. We follow the experimental settings in (Diao et al., 2023) to simulate realistic few-shot learning scenarios. Specifically, we randomly sample $\zeta$ examples from each class in the original training data to construct the training set and use a separate set of $\zeta$ examples for the development set. The original development set is designated as the test set. Accuracy is employed as the evaluation metric across all datasets.

**Baselines.** We consider the following black-box prompt learning methods as our baselines:

- **Manual Prompt (Manual):** directly conducts zero-shot evaluations on pre-trained, fixed LLMs without engaging in any additional learning or fine-tuning processes.
- **GAP3:** leverages additional LLMs to generate prompts from an empty template and employs a genetic algorithm to select the most effective prompts (Zhao et al., 2023).
- **BBT:** projects the original space onto a subspace via a random matrix, after which the prompt is optimized within this reduced-dimensional space (Sun et al., 2022b).
- **SSPT:** extends the BBT optimization paradigm by incorporating subspace learning and selection techniques to identify the optimal ultra-low-dimensional subspace, thereby replacing the previously utilized random subspace (Zhang et al., 2024).
- **BDPL:** frames the prompt learning problem as a distributed optimization task and optimizes it using policy gradient methods (Diao et al., 2023).

**Implementation Details.** We implement our code [1] using Python 3.9 and PyTorch 2.4, conducting experiments primarily on a computing cluster with NVIDIA A40 GPUs. Detailed information re-

---

[1]The vocabulary $\mathcal{V}$ is constructed following Diao et al. (2023)

garding the hyperparameters and templates used in the experiments can be found in the Appendix A.7. Our code is available at the following URL: https://anonymous.4open.science/r/LEAP.

## 5.2 MAIN RESULTS

We use RoBERTa-large Liu et al. (2019), GPT2-XL Radford et al. (2019), and Llama3 AI@Meta (2024) as our primary backbone black-box LLMs. These models comprise approximately 355 million, 1.5 billion, and 8 billion parameters, respectively. The weights of the pre-trained models are obtained from Hugging Face. To assess the effectiveness of our proposed approach, we compare it against baseline methods under prompt length configurations of 20 and 50 tokens. Since the baselines cannot effectively compute the objective function where the label words are missing, we employ the interaction LLM to generate usable label words for them. Specifically, we first divide the training set by category and then input each category in batches into the LLM. Finally, we count the occurrences of the most probable tokens in the model's token vocabulary outputs for each category and select the token with the highest count as the label word for the corresponding category's data. The text classification accuracy results of LEAP and baselines are reported in Table 1-Table 3. Each result is based on three Monte Carlo experiments. It can be seen that our approach shows a clear advantage compared to all the prompt learning baselines. For example, on the SST-2 dataset, LEAP achieves an accuracy of 78.40% for the RoBERTa-large model using a 20-length prompt, which is notably higher than the second-best method, BBT, at 61.28%. In the setting of the prompt length is 50, LEAP maintains its superiority. We include an intuitive display of the prompt words learned by our method in Table 4. More results are given in Appendix A.8.

Table 1: Comparison results of the four baseline methods and our method (LEAP) on RoBERTa-large in the percentage of average text classification accuracy $\pm$ standard deviation.

| Length | Method | BOOK | CoLA | ELEC | QNLI | RTE | SNLI | SST-2 | AG |
|---|---|---|---|---|---|---|---|---|---|
| - | Manual | $94.47_{\pm1.67}$ | $50.91_{\pm3.16}$ | $71.67_{\pm27.01}$ | $50.27_{\pm0.00}$ | $47.17_{\pm5.60}$ | $36.00_{\pm0.00}$ | $53.52_{\pm8.39}$ | $35.41_{\pm6.10}$ |
| 20 | GAP3 | $90.83_{\pm0.96}$ | $55.67_{\pm19.65}$ | $41.63_{\pm35.19}$ | $49.58_{\pm0.16}$ | $48.62_{\pm3.62}$ | $32.90_{\pm0.08}$ | $49.16_{\pm1.53}$ | $25.12_{\pm0.55}$ |
| | BBT | $94.40_{\pm1.82}$ | $53.18_{\pm3.46}$ | $71.13_{\pm28.58}$ | $50.10_{\pm1.00}$ | $53.43_{\pm3.75}$ | $37.02_{\pm0.50}$ | $61.28_{\pm15.38}$ | $36.75_{\pm6.43}$ |
| | SSPT | $94.33_{\pm1.88}$ | $45.32_{\pm12.32}$ | $68.56_{\pm32.97}$ | $48.26_{\pm0.12}$ | $51.50_{\pm5.42}$ | $34.43_{\pm0.99}$ | $58.56_{\pm16.53}$ | $37.71_{\pm5.21}$ |
| | BDPL | $94.23_{\pm2.25}$ | $55.48_{\pm10.12}$ | $72.15_{\pm22.50}$ | $48.64_{\pm1.58}$ | $49.10_{\pm4.33}$ | $34.94_{\pm0.13}$ | $59.94_{\pm13.56}$ | $37.47_{\pm6.06}$ |
| | LEAP | $\mathbf{95.23}_{\pm0.87}$ | $\mathbf{56.34}_{\pm22.14}$ | $\mathbf{92.97}_{\pm0.70}$ | $\mathbf{51.14}_{\pm0.87}$ | $\mathbf{54.27}_{\pm3.98}$ | $\mathbf{37.16}_{\pm1.84}$ | $\mathbf{78.40}_{\pm11.53}$ | $\mathbf{55.01}_{\pm9.12}$ |
| 50 | GAP3 | $90.83_{\pm0.96}$ | $55.67_{\pm19.65}$ | $41.63_{\pm35.19}$ | $49.56_{\pm0.16}$ | $46.57_{\pm4.51}$ | $32.90_{\pm0.08}$ | $49.16_{\pm1.53}$ | $25.12_{\pm0.55}$ |
| | BBT | $94.30_{\pm2.27}$ | $53.40_{\pm4.17}$ | $65.10_{\pm27.51}$ | $50.42_{\pm0.51}$ | $52.35_{\pm2.01}$ | $37.73_{\pm1.85}$ | $64.18_{\pm11.34}$ | $37.75_{\pm5.34}$ |
| | SSPT | $94.30_{\pm2.36}$ | $43.94_{\pm0.55}$ | $69.00_{\pm31.32}$ | $49.66_{\pm0.70}$ | $49.70_{\pm4.64}$ | $35.43_{\pm2.73}$ | $52.10_{\pm9.89}$ | $37.65_{\pm4.73}$ |
| | BDPL | $94.43_{\pm2.24}$ | $55.19_{\pm8.60}$ | $70.32_{\pm27.62}$ | $49.42_{\pm1.78}$ | $51.38_{\pm2.21}$ | $33.90_{\pm2.28}$ | $60.82_{\pm17.07}$ | $37.99_{\pm3.71}$ |
| | LEAP | $\mathbf{95.43}_{\pm1.07}$ | $\mathbf{56.34}_{\pm22.14}$ | $\mathbf{93.53}_{\pm0.52}$ | $\mathbf{50.79}_{\pm0.29}$ | $\mathbf{52.83}_{\pm3.62}$ | $\mathbf{37.83}_{\pm1.17}$ | $\mathbf{84.82}_{\pm2.31}$ | $\mathbf{56.20}_{\pm8.09}$ |

Table 2: Comparison results of the four baseline methods and our method (LEAP) on GPT2-XL in the percentage of average text classification accuracy $\pm$ standard deviation.

| Length | Method | BOOK | CoLA | ELEC | QNLI | RTE | SNLI | SST-2 | AG |
|---|---|---|---|---|---|---|---|---|---|
| - | Manual | $53.27_{\pm10.71}$ | $53.60_{\pm11.88}$ | $63.29_{\pm0.00}$ | $49.47_{\pm1.85}$ | $49.82_{\pm0.00}$ | $33.78_{\pm1.12}$ | $55.62_{\pm3.64}$ | $25.39_{\pm0.38}$ |
| 20 | GAP3 | $38.55_{\pm17.63}$ | $43.59_{\pm21.78}$ | $61.31_{\pm0.53}$ | $50.18_{\pm0.62}$ | $47.29_{\pm0.00}$ | $33.95_{\pm0.77}$ | $52.65_{\pm2.85}$ | $25.09_{\pm0.25}$ |
| | BBT | $38.43_{\pm46.31}$ | $55.67_{\pm21.07}$ | $13.43_{\pm0.63}$ | $50.18_{\pm0.62}$ | $47.29_{\pm0.00}$ | $33.12_{\pm0.23}$ | $51.61_{\pm2.70}$ | $25.01_{\pm0.01}$ |
| | SSPT | $40.13_{\pm44.83}$ | $56.38_{\pm15.71}$ | $18.88_{\pm0.23}$ | $50.18_{\pm0.62}$ | $47.28_{\pm0.00}$ | $33.07_{\pm0.23}$ | $48.47_{\pm2.73}$ | $26.25_{\pm2.69}$ |
| | BDPL | $35.53_{\pm8.47}$ | $49.60_{\pm13.61}$ | $36.56_{\pm24.29}$ | $49.87_{\pm0.45}$ | $45.85_{\pm1.44}$ | $33.03_{\pm0.68}$ | $54.89_{\pm3.32}$ | $25.21_{\pm0.79}$ |
| | LEAP | $\mathbf{42.93}_{\pm6.12}$ | $\mathbf{57.56}_{\pm20.04}$ | $\mathbf{71.61}_{\pm19.59}$ | $\mathbf{50.21}_{\pm0.00}$ | $\mathbf{54.75}_{\pm2.92}$ | $\mathbf{35.24}_{\pm0.62}$ | $\mathbf{56.00}_{\pm11.98}$ | $\mathbf{61.00}_{\pm7.01}$ |
| 50 | GAP3 | $38.55_{\pm17.63}$ | $43.59_{\pm21.78}$ | $61.31_{\pm0.53}$ | $50.18_{\pm0.62}$ | $47.29_{\pm0.00}$ | $33.95_{\pm0.77}$ | $52.65_{\pm2.85}$ | $25.09_{\pm0.25}$ |
| | BBT | $39.07_{\pm45.76}$ | $56.22_{\pm21.29}$ | $13.39_{\pm0.30}$ | $50.18_{\pm0.62}$ | $47.29_{\pm0.00}$ | $33.13_{\pm0.23}$ | $50.54_{\pm1.78}$ | $24.98_{\pm0.08}$ |
| | SSPT | $40.70_{\pm44.25}$ | $56.38_{\pm17.20}$ | $20.84_{\pm0.00}$ | $50.18_{\pm0.62}$ | $47.29_{\pm0.00}$ | $32.99_{\pm0.28}$ | $48.81_{\pm2.25}$ | $25.87_{\pm2.87}$ |
| | BDPL | $31.17_{\pm11.28}$ | $57.11_{\pm9.21}$ | $32.72_{\pm20.07}$ | $49.92_{\pm1.08}$ | $46.69_{\pm2.66}$ | $33.54_{\pm0.38}$ | $51.95_{\pm0.98}$ | $25.41_{\pm0.80}$ |
| | LEAP | $\mathbf{49.17}_{\pm18.93}$ | $\mathbf{58.36}_{\pm18.65}$ | $\mathbf{65.01}_{\pm37.56}$ | $\mathbf{50.21}_{\pm0.01}$ | $\mathbf{53.07}_{\pm2.17}$ | $\mathbf{33.98}_{\pm0.04}$ | $\mathbf{52.68}_{\pm6.22}$ | $\mathbf{60.34}_{\pm12.42}$ |

## 5.3 ABLATION STUDY

In our method, we utilize two core techniques—$\ell_1$-norm and Gumbel-Softmax—to optimize the prompts and the $M$ matrix. To further demonstrate the effectiveness of these mechanisms, we conduct an ablation study. The experimental results are presented in Figure 2. It is evident that both $\ell_1$-norm and Gumbel-Softmax positively influence our approach.

Table 3: Comparison results of the four baseline methods and our method (LEAP) on Llama3 in the percentage of average text classification accuracy $\pm$ standard deviation.

| Length | Method | BOOK | CoLA | ELEC | QNLI | RTE | SNLI | SST-2 | AG |
|---|---|---|---|---|---|---|---|---|---|
| - | Manual | $49.30_{\pm 33.26}$ | $47.75_{\pm 16.34}$ | $73.94_{\pm 16.54}$ | $49.95_{\pm 0.00}$ | $53.67_{\pm 0.83}$ | $33.73_{\pm 1.72}$ | $48.20_{\pm 5.36}$ | $28.02_{\pm 2.68}$ |
| 20 | GAP3 | $60.21_{\pm 31.02}$ | $50.11_{\pm 16.59}$ | $75.57_{\pm 18.84}$ | $50.54_{\pm 0.00}$ | $47.29_{\pm 0.00}$ | $32.87_{\pm 0.00}$ | $49.12_{\pm 4.52}$ | $25.28_{\pm 8.15}$ |
| | BBT | $28.40_{\pm 16.32}$ | $48.96_{\pm 16.40}$ | $67.62_{\pm 34.20}$ | $50.24_{\pm 0.35}$ | $47.77_{\pm 2.18}$ | $33.92_{\pm 1.39}$ | $49.24_{\pm 1.84}$ | $27.77_{\pm 2.09}$ |
| | SSPT | $26.93_{\pm 15.59}$ | $52.09_{\pm 17.30}$ | $64.18_{\pm 40.99}$ | $50.54_{\pm 0.00}$ | $47.29_{\pm 0.00}$ | $33.51_{\pm 0.55}$ | $49.04_{\pm 2.45}$ | $25.64_{\pm 1.52}$ |
| | BDPL | $33.13_{\pm 5.28}$ | $56.15_{\pm 6.53}$ | $62.20_{\pm 17.93}$ | $50.58_{\pm 0.46}$ | $52.35_{\pm 2.37}$ | $33.76_{\pm 0.33}$ | $48.17_{\pm 5.98}$ | $30.59_{\pm 2.93}$ |
| | LEAP | $\mathbf{61.60}_{\pm 19.26}$ | $\mathbf{60.91}_{\pm 13.90}$ | $\mathbf{76.64}_{\pm 10.75}$ | $\mathbf{50.78}_{\pm 2.27}$ | $\mathbf{53.07}_{\pm 1.66}$ | $\mathbf{35.89}_{\pm 1.43}$ | $\mathbf{53.86}_{\pm 8.28}$ | $\mathbf{69.93}_{\pm 5.76}$ |
| 50 | GAP3 | $60.21_{\pm 31.02}$ | $50.11_{\pm 16.59}$ | $75.57_{\pm 18.84}$ | $50.54_{\pm 0.00}$ | $47.29_{\pm 0.00}$ | $32.87_{\pm 0.00}$ | $49.12_{\pm 4.52}$ | $25.28_{\pm 8.15}$ |
| | BBT | $42.60_{\pm 29.68}$ | $53.08_{\pm 15.88}$ | $69.56_{\pm 5.49}$ | $48.48_{\pm 2.29}$ | $47.53_{\pm 3.24}$ | $33.92_{\pm 1.39}$ | $49.58_{\pm 0.43}$ | $28.00_{\pm 1.82}$ |
| | SSPT | $34.90_{\pm 23.73}$ | $52.92_{\pm 16.95}$ | $70.52_{\pm 30.64}$ | $50.54_{\pm 0.01}$ | $47.29_{\pm 0.00}$ | $33.51_{\pm 0.55}$ | $51.11_{\pm 2.47}$ | $26.76_{\pm 1.46}$ |
| | BDPL | $40.40_{\pm 11.36}$ | $52.16_{\pm 5.64}$ | $74.83_{\pm 13.61}$ | $50.51_{\pm 0.69}$ | $51.50_{\pm 2.12}$ | $33.99_{\pm 0.80}$ | $48.43_{\pm 4.44}$ | $30.05_{\pm 4.07}$ |
| | LEAP | $\mathbf{75.60}_{\pm 19.29}$ | $\mathbf{57.30}_{\pm 19.99}$ | $\mathbf{80.60}_{\pm 14.11}$ | $\mathbf{50.69}_{\pm 2.12}$ | $\mathbf{51.62}_{\pm 2.87}$ | $\mathbf{34.28}_{\pm 1.68}$ | $\mathbf{52.33}_{\pm 5.63}$ | $\mathbf{61.91}_{\pm 8.41}$ |

Table 4: Example prompts of our method on SST-2. $\times$ denotes the samples that are incorrectly predicted, while $\checkmark$ denotes those that are correctly predicted after applying the learned prompts.

| Model | Prompt+Sentence | Prediction |
|---|---|---|
| | The turkey would've been a far better title. | $\times$ |
| RoBERTa | only new been an enough a action more us enough and good movies by what he up to a own The cold turkey would've been a far better title. | $\checkmark$ |
| GPT2-XL | makes on we enough this little your just the from he your out he are are or be he their   The turkey would've been a far better title. | $\checkmark$ |
| Llama3 | but make made that by own no great as one humor time will most for about their are your who The turkey would've been a far better title. | $\checkmark$ |

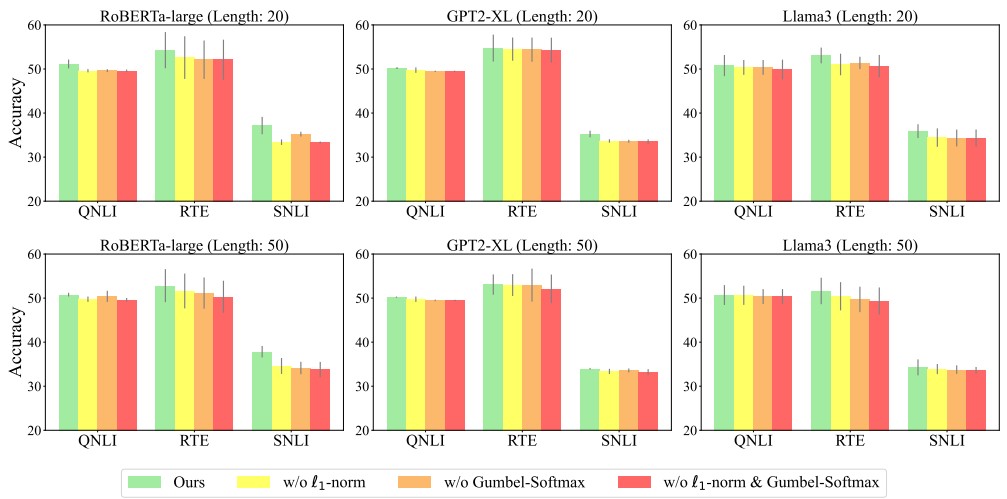

Figure 2: Ablations of the using components, $\ell_1$-norm and Gumbel-Softmax, on RoBERTa-Large, GPT2-XL, and Llama3 with the prompt lengths of 20 (top) and 50 (bottom), respectively.

## 6 CONCLUSION

In this paper, we propose LEAP, a novel solution to the critical challenge of black-box prompt learning within the context of LMaaS, particularly in scenarios where label vocabulary is missing. Our method employs an alternating optimization framework to jointly learn discrete prompt tokens and a mapping matrix that converts the full token vocabulary outputs of LLMs into task-specific categories. Notably, LEAP is the first work to effectively learn discrete prompts without relying on a predefined label vocabulary. Theoretical analysis confirms the convergence of our proposed algorithm under standard assumptions, ensuring its reliability. Extensive evaluations across various LLMs demonstrate the superior performance of our approach.

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

# A APPENDIX

## A.1 NOTATIONS

**Symbols**

| | |
|---|---|
| $(\cdot, \cdot, \cdot)$ | Row vector |
| $(\cdot; \cdot; \cdot)$ | Column vector |
| $(\cdot)^{\top}$ | Transpose operation |
| $\|\cdot\|_1$ | $\ell_1$-norm |
| $\|\cdot\|_2$ | $\ell_2$-norm |
| $\sum_{\{\phi_i \sim \boldsymbol{p}_i\}_{i=1}^n}$ | Abbreviation for $\sum_{\phi_1 \sim \boldsymbol{p}_1} \cdots \sum_{\phi_n \sim \boldsymbol{p}_n}$ |
| $\mathbb{E}_{\{\Phi^r\}_{r=1}^{I_{\boldsymbol{\alpha}}}}$ | Abbreviation for $\mathbb{E}_{\{\Phi^r \sim \mathcal{S}(\alpha)\}_{r=1}^{I_{\boldsymbol{\alpha}}}}$ |
| $\mathbb{E}_{\{\Phi^s\}_{s=1}^{I_M}}$ | Abbreviation for $\mathbb{E}_{\{\Phi^s \sim \mathcal{S}(\alpha)\}_{s=1}^{I_M}}$ |
| $\mathbb{E}_{(\boldsymbol{x}_k, \boldsymbol{y}_k) \in \mathcal{D}}$ | Expectation for the $k$-th sample in $\mathcal{D}$ |
| $\mathbb{E}_{\mathcal{B}}$ | Expectation for the mini-batch |
| $\mathbb{E}_{\{\Phi^r \sim \mathcal{S}(\boldsymbol{\alpha})\}_{r=1}^{I_{\boldsymbol{\alpha}}}}$ | Expectation for prompt sampling when updating $\Phi$ |
| $\mathbb{E}_{\{\Phi^s \sim \mathcal{S}(\boldsymbol{\alpha})\}_{s=1}^{I_M}}$ | Expectation for prompt sampling when updating $M$ |

**Variables**

| | |
|---|---|
| $\mathcal{D} = \{\boldsymbol{X}, \boldsymbol{Y}\}$ | Training dataset |
| $\boldsymbol{X} = \{\boldsymbol{x}_k\}_{k=1}^K$ | Input sentences in $\mathcal{D}$ |
| $\boldsymbol{Y} = \{\boldsymbol{y}_k\}_{k=1}^K$ | One-hot encoded vectors in $\mathcal{D}$ |
| $\boldsymbol{x}_k$ | $k$-th input tokens vector |
| $\boldsymbol{y}_k$ | $k$-th one-hot encoded vector |
| $\mathcal{B} = \{\boldsymbol{x}_k, \boldsymbol{y}_k\}_{k=1}^B$ | Mini-batch |
| $n$ | Prompt length |
| $\Phi = \phi_1 ... \phi_i ... \phi_n$ | Discrete prompt |
| $\phi_i$ | $i$-th prompt token |
| $N$ | Vocabulary size |
| $\mathcal{V} = (\mathcal{V}[1], ... \mathcal{V}[j], ... \mathcal{V}[N])$ | Prompt vocabulary |
| $\boldsymbol{P} = (\boldsymbol{p}_1; ... \boldsymbol{p}_i; ... \boldsymbol{p}_n)$ | Prompt probability matrix |
| $\boldsymbol{p}_i = (p_{i,1}, ... p_{i,j}, ... p_{i,N})$ | Prompt probability vector for the $i$-th token |
| $\boldsymbol{\alpha} = (\alpha_{i,j})_{n \times N}$ | Learnable parameter in Gumbel-Softmax |
| $\tau$ | Temperature parameter in Gumbel-Softmax |
| $I_{\boldsymbol{\alpha}}$ | Sampling times of prompt when updating $\boldsymbol{\alpha}$ |
| $\eta_{\boldsymbol{\alpha}}$ | Learning rate when updating $\boldsymbol{\alpha}$ |
| $D$ | Length of LLM's vocabulary |
| $C$ | Number of categories |
| $\boldsymbol{M} = (m_{d,c})_{D \times C}$ | Mapping matrix |
| $I_M$ | Sampling times of prompt when updating $M$ |
| $\eta_M$ | Learning rate when updating $M$ |

## Functions

| | |
|---|---|
| $\mathrm{Cat}(\cdot)$ | Categorical distribution |
| $\Phi \sim \mathcal{S}(\boldsymbol{\alpha})$ | Abbreviation for $\Phi \sim \mathrm{Cat}(\mathcal{S}(\boldsymbol{\alpha}))$ |
| $\mathrm{Softmax}(\cdot)$ | Softmax function |
| $\mathcal{S}(\cdot)$ | Gumbel-Softmax function |
| $\mathcal{G}(\cdot, \cdot)$ | LLM model |
| $\mathcal{L}(\cdot, \cdot)$ | Cross entropy loss function |
| $\mathcal{L}(\Phi, \boldsymbol{M}; \boldsymbol{d}_k)$ | Abbreviation for $\mathcal{L}(\mathcal{G}(\Phi, \boldsymbol{d}_k) \cdot \boldsymbol{M}, \boldsymbol{y}_k)$ |
| $f_{\mathcal{D}}(\boldsymbol{\alpha}, \boldsymbol{M})$ | Abbreviation for $\frac{1}{K} \sum_{(\boldsymbol{x}_k, \boldsymbol{y}_k) \in \mathcal{D}} \mathbb{E}_{\Phi \sim \mathcal{S}(\boldsymbol{\alpha})} \left[ \mathcal{L}(\mathcal{G}(\Phi, \boldsymbol{x}_k) \cdot \boldsymbol{M}, \boldsymbol{y}_k) \right]$ |
| $f_{\mathcal{B}}(\boldsymbol{\alpha}, \boldsymbol{M})$ | Abbreviation for $\frac{1}{B} \sum_{(\boldsymbol{x}_k, \boldsymbol{y}_k) \in \mathcal{B}} \mathbb{E}_{\Phi \sim \mathcal{S}(\boldsymbol{\alpha})} \left[ \mathcal{L}(\mathcal{G}(\Phi, \boldsymbol{x}_k) \cdot \boldsymbol{M}, \boldsymbol{y}_k) \right]$ |
| $f_k(\boldsymbol{\alpha}, \boldsymbol{M})$ | Abbreviation for $\mathbb{E}_{\Phi \sim \mathcal{S}(\boldsymbol{\alpha})} \left[ \mathcal{L}(\mathcal{G}(\Phi, \boldsymbol{x}_k) \cdot \boldsymbol{M}, \boldsymbol{y}_k) \right]$ |

## Gradients

| | |
|---|---|
| $\nabla_{\boldsymbol{\alpha}_i} f_{\mathcal{D}}(\cdot, \cdot)$ | Full gradient for $\boldsymbol{\alpha}_i$ |
| $\nabla_{\boldsymbol{\alpha}_i} f_k(\cdot, \cdot)$ | Stochastic gradient for $\boldsymbol{\alpha}_i$ |
| $\nabla_{\boldsymbol{\alpha}_i} f_{\mathcal{B}}(\cdot, \cdot)$ | Mini-batch stochastic gradient for $\boldsymbol{\alpha}_i$ |
| $\hat{\nabla}_{\boldsymbol{\alpha}_i} f(\cdot, \cdot)$ | Variance-reduced policy gradient for $\boldsymbol{\alpha}_i$ |
| $\nabla_{\boldsymbol{M}} f_{\mathcal{D}}(\cdot, \cdot)$ | Full gradient for $\boldsymbol{M}$ |
| $\nabla_{\boldsymbol{M}} f_k(\cdot, \cdot)$ | Stochastic gradient for $\boldsymbol{M}$ |
| $\nabla_{\boldsymbol{M}} f_{\mathcal{B}}(\cdot, \cdot)$ | Mini-batch stochastic gradient for $\boldsymbol{M}$ |
| $\tilde{\nabla}_{\boldsymbol{M}} f(\cdot, \cdot)$ | Gradient with prompt sampling for $\boldsymbol{M}$ |
| $\tilde{g}(\cdot, \cdot)$ | Gradient mapping with prompt sampling for $\boldsymbol{M}$ |
| $g_{\mathcal{B}}(\cdot, \cdot)$ | Gradient mapping of mini-batch stochastic gradient for $\boldsymbol{M}$ |
| $g_{\mathcal{D}}(\cdot, \cdot)$ | Gradient mapping of full gradient for $\boldsymbol{M}$ |

### A.2 THE DERIVATIVE PROCESS FOR GUMBEL-SOFTMAX FUNCTION

$$
\frac{\partial \log p_{i,j_i}}{\partial \alpha_{i,j}}
$$

$$
= \frac{\partial}{\partial \alpha_{i,j}} \left( \log \left( \frac{\exp\left(\frac{\log(\alpha_{i,j_i}) + g_{i,j_i}}{\tau}\right)}{\sum_{\rho=1}^{N} \exp\left(\frac{\log(\alpha_{i,\rho}) + g_{i,\rho}}{\tau}\right)} \right) \right)
$$

$$
= \frac{\partial}{\partial \alpha_{i,j}} \left( \log \left( \exp\left(\frac{\log(\alpha_{i,j_i}) + g_{i,j_i}}{\tau}\right) \right) \right) - \frac{\partial}{\partial \alpha_{i,j}} \left( \log \left( \sum_{\rho=1}^{N} \exp\left(\frac{\log(\alpha_{i,\rho}) + g_{i,\rho}}{\tau}\right) \right) \right).
$$

$$(19)$$

According to the derivation rule of the Softmax function,
when $j = j_i$:

$$
\frac{\partial p_{i,j_i}}{\partial \alpha_{i,j_i}} = \frac{1}{\tau \alpha_{i,j_i}} - p_{i,j_i} \cdot \frac{1}{\tau \alpha_{i,j_i}} = \frac{1 - p_{i,j_i}}{\tau \alpha_{i,j_i}}. \tag{20}
$$

when $j \neq j_i$:

$$
\frac{\partial p_{i,j_i}}{\partial \alpha_{i,j}} = -\frac{p_{i,j}}{\tau \alpha_{i,j}}. \tag{21}
$$

### A.3 Gradient mapping functions for M

We define the gradient mapping functions as follows (J Reddi et al., 2016, Eq. (5)):

$$\tilde{g}_{\mathcal{B}}\left(\boldsymbol{\alpha}_{t+1}, \boldsymbol{M}_t\right) = \frac{1}{\eta_M}\left(\boldsymbol{M}_t - \text{prox}_{\eta_M r}\left[\boldsymbol{M}_t - \eta_M \cdot \tilde{\nabla}_{\boldsymbol{M}} f_{\mathcal{B}}\left(\boldsymbol{\alpha}_{t+1}, \boldsymbol{M}_t\right)\right]\right), \qquad (22)$$

$$g_{\mathcal{B}}\left(\boldsymbol{\alpha}_{t+1}, \boldsymbol{M}_t\right) = \frac{1}{\eta_M}\left(\boldsymbol{M}_t - \text{prox}_{\eta_M r}\left[\boldsymbol{M}_t - \eta_M \cdot \nabla_{\boldsymbol{M}} f_{\mathcal{B}}\left(\boldsymbol{\alpha}_{t+1}, \boldsymbol{M}_t\right)\right]\right), \qquad (23)$$

$$g_{\mathcal{D}}\left(\boldsymbol{\alpha}_{t+1}, \boldsymbol{M}_t\right) = \frac{1}{\eta_M}\left(\boldsymbol{M}_t - \text{prox}_{\eta_M r}\left[\boldsymbol{M}_t - \eta_M \cdot \nabla_{\boldsymbol{M}} f_{\mathcal{D}}\left(\boldsymbol{\alpha}_{t+1}, \boldsymbol{M}_t\right)\right]\right). \qquad (24)$$

Consequently, when the learning rate is set to $\eta_M$, the update of $\boldsymbol{M}$ can be reformulated as follows:

$$\boldsymbol{M}_{t+1} = \boldsymbol{M}_t - \eta_M \cdot \tilde{g}_{\mathcal{B}}\left(\boldsymbol{\alpha}_{t+1}, \boldsymbol{M}_t\right). \qquad (25)$$

Additionally, we adopt the gradient mapping $g_{\mathcal{D}}\left(\boldsymbol{\alpha}, \boldsymbol{M}\right)$ as the convergence criterion for $\boldsymbol{M}$ in this study (Consistent with (Li & Li, 2018; Ghadimi & Lan, 2013)).

### A.4 Assumptions and Lemmas

**Assumption 1** (Bounded variance of stochastic gradients). *The stochastic gradients is unbiased and we assume the variance of stochastic gradients for $\boldsymbol{\alpha}_i$ and $\boldsymbol{M}$ is bounded:*

$$\mathbb{E}_{(\boldsymbol{x}_k, \boldsymbol{y}_k) \in \mathcal{D}} \left\| \nabla_{\boldsymbol{\alpha}_i} f_k(\boldsymbol{\alpha}, \boldsymbol{M}) - \mathbb{E}_{(\boldsymbol{x}_k, \boldsymbol{y}_k) \in \mathcal{D}} \left[\nabla_{\boldsymbol{\alpha}_i} f_k(\boldsymbol{\alpha}, \boldsymbol{M})\right] \right\|_2^2 \leq \sigma_{\boldsymbol{\alpha}}^2; \qquad (26)$$

$$\mathbb{E}_{(\boldsymbol{x}_k, \boldsymbol{y}_k) \in \mathcal{D}} \left\| \nabla_{\boldsymbol{M}} f_k(\boldsymbol{\alpha}, \boldsymbol{M}) - \mathbb{E}_{(\boldsymbol{x}_k, \boldsymbol{y}_k) \in \mathcal{D}} \left[\nabla_{\boldsymbol{M}} f_k(\boldsymbol{\alpha}, \boldsymbol{M})\right] \right\|_2^2 \leq \sigma_{\boldsymbol{M}}^2. \qquad (27)$$

**Assumption 2** (Lower Boundedness for objective function). *Given an initial point $(\boldsymbol{\alpha}_0, \boldsymbol{M}_0)$, $(\boldsymbol{\alpha}_*, \boldsymbol{M}_*)$ denotes the global minimum of $F(\boldsymbol{\alpha}, \boldsymbol{M}; \mathcal{D})$, there exists $\triangle < \infty$ such that*

$$F(\boldsymbol{\alpha}_0, \boldsymbol{M}_0; \mathcal{D}) - F(\boldsymbol{\alpha}_*, \boldsymbol{M}_*; \mathcal{D}) \leq \triangle. \qquad (28)$$

**Assumption 3** (Bounded Loss). *We perform a clipping operation with a constant $G$ for loss function:*

$$|\mathcal{L}(\Phi, \boldsymbol{M}; \mathcal{D})| \leq U. \qquad (29)$$

**Assumptions 1 and 2** constitute the foundational premises for addressing non-convex optimization problems using stochastic gradient descent, as demonstrated in prior studies (Ghadimi & Lan, 2013; Hazan & Kale, 2014; Xu et al., 2019; Liu et al., 2020). **Assumption 3** ensures that the loss function remains bounded by regulating the loss during the estimation of the $I_{\boldsymbol{\alpha}}$-th and $I_{\boldsymbol{M}}$-th samples when updating $\boldsymbol{\alpha}$ and $\boldsymbol{M}$. This boundedness is essential for facilitating rigorous theoretical analysis. It is important to recognize that loss functions, such as the cross-entropy function, can potentially become unbounded. In practical applications, these loss values are typically clipped to maintain boundedness.

The following **Lemma 1** and **2** show that the $f_{\mathcal{D}}(\boldsymbol{\alpha}, \boldsymbol{M})$ is lipschitz smooth for $\boldsymbol{\alpha}$ and $\boldsymbol{M}$, the **Lemma 3** and **4** show that the unbiasedness and bounded variance of prompt sampling of $f_{\mathcal{D}}(\boldsymbol{\alpha}, \boldsymbol{M})$ for $\boldsymbol{\alpha}$ and $\boldsymbol{M}$. These lemmas are important for convergence analysis of LEAP.

**Lemma 1** (Lipschitz smoothness for $\boldsymbol{\alpha}$). *Let $\alpha_{i,j} \geq \beta > 0$ for $i = 1, ..., n$ and $j = 1, ..., N$, $\tau > 0$ is the temperature parameter, the full loss function $f_{\mathcal{D}}(\boldsymbol{\alpha}, \boldsymbol{M})$ is lipschitz smooth for $\boldsymbol{\alpha}$ with smooth constant $L_{\boldsymbol{\alpha}} = \frac{nUN(\tau+1)}{\tau^2 \beta^2}$.*

*Proof.* We can compute the Hessian of the full loss function (3) for $\boldsymbol{\alpha}$, $\forall i', i'' \in 1, \cdots, n$ and $j', j'' \in 1, \cdots, N$:
1) if $i' \neq i''$, we process $\phi_{i'}$ and $\phi_{i''}$:

$$\frac{\partial^2}{\partial \alpha_{i',j'} \partial \alpha_{i'',j''}} f_{\mathcal{D}}(\boldsymbol{\alpha}, \boldsymbol{M})$$

$$= \sum_{\substack{\{\phi_i \sim \boldsymbol{p}_i\}_{i=1}^n \\ i \neq i', i''}} \left( \sum_{\phi_{i'} \sim \boldsymbol{p}_{i'}} \sum_{\phi_{i''} \sim \boldsymbol{p}_{i''}} \left( \mathcal{L}(\Phi, \boldsymbol{M}; \mathcal{D}) \cdot \frac{\partial^2 \prod_{i=1}^n \mathcal{P}(\phi_i)}{\partial \alpha_{i',j'} \partial \alpha_{i'',j''}} \right) \right)$$

$$= \sum_{\substack{\{\phi_i \sim \boldsymbol{p}_i\}_{i=1}^n \\ i \neq i', i''}} \left( \mathcal{L}\left(\Phi, \boldsymbol{M}; \mathcal{D}\right) \cdot \frac{\partial^2 \left[\mathcal{P}(\phi_{i'})\mathcal{P}(\phi_{i''})\right]}{\partial \alpha_{i',j'} \partial \alpha_{i'',j''}} \cdot \prod_{\substack{i=1 \\ i \neq i', i''}}^n \mathcal{P}(\phi_i) \right).$$

We compute the second order partial derivative of $\partial^2 \left[\mathcal{P}(\phi_{i'})\mathcal{P}(\phi_{i''})\right]$:

$$\frac{\partial^2 \left[\mathcal{P}(\phi_{i'})\mathcal{P}(\phi_{i''})\right]}{\partial \alpha_{i',j'} \partial \alpha_{i'',j''}} = \begin{cases} \frac{1-p_{i',j_i'}}{\tau \alpha_{i',j_i'}} \cdot \frac{1-p_{i'',j_i''}}{\tau \alpha_{i'',j_i''}}, & \text{if } j' = j_i' \text{ and } j'' = j_i''; \\ \frac{1-p_{i',j_i'}}{\tau \alpha_{i',j_i'}} \cdot \left( -\frac{p_{i'',j''}}{\tau \alpha_{i'',j''}} \right), & \text{if } j' = j_i' \text{ and } j'' \neq j_i''; \\ -\frac{p_{i',j'}}{\tau \alpha_{i',j'}} \cdot \frac{1-p_{i'',j_i''}}{\tau \alpha_{i'',j_i''}}, & \text{if } j' \neq j_i' \text{ and } j'' = j_i''; \\ -\frac{p_{i',j'}}{\tau \alpha_{i',j'}} \cdot \left( -\frac{p_{i'',j''}}{\tau \alpha_{i'',j''}} \right), & \text{if } j' \neq j_i' \text{ and } j'' \neq j_i''. \end{cases}$$

Then, based on $\alpha_{i,j} \geq \beta > 0$ and **Assumption 3**, the second-order partial derivative of $f_\mathcal{D}(\boldsymbol{\alpha}, \boldsymbol{M})$ can be bounded:

$$\left| \frac{\partial^2}{\partial \alpha_{i',j'} \partial \alpha_{i'',j''}} \left[f_\mathcal{D}(\boldsymbol{\alpha}, \boldsymbol{M})\right] \right|$$

$$\leq \sum_{\substack{\{\phi_i \sim \boldsymbol{p}_i\}_{i=1}^n \\ i \neq i', i''}} \left( \left| \mathcal{L}\left(\Phi, \boldsymbol{M}; \mathcal{D}\right) \right| \cdot \prod_{\substack{i=1 \\ i \neq i', i''}}^n \mathcal{P}(\phi_i) \right) \cdot \left| \frac{\partial^2 \left[\mathcal{P}(\phi_{i'})\mathcal{P}(\phi_{i''})\right]}{\partial \alpha_{i',j'} \partial \alpha_{i'',j''}} \right|$$

$$\leq U \cdot \frac{1}{\tau^2 \beta^2} = \frac{U}{\tau^2 \beta^2}.$$

2) If $i' = i''$, we process $\phi_{i'}$:

$$\frac{\partial^2}{\partial \alpha_{i',j'} \partial \alpha_{i',j''}} \left[f_\mathcal{D}(\boldsymbol{\alpha}, \boldsymbol{M})\right] = \sum_{\substack{\{\phi_i \sim \boldsymbol{p}_i\}_{i=1}^n \\ i \neq i'}} \left( \mathcal{L}\left(\Phi, \boldsymbol{M}; \mathcal{D}\right) \frac{\partial^2 \left[\mathcal{P}(\phi_{i'})\right]}{\partial \alpha_{i',j'} \partial \alpha_{i',j''}} \prod_{\substack{i=1 \\ i \neq i'}}^n \mathcal{P}(\phi_i) \right).$$

Similar to the analysis in case $i' \neq i''$, we can get:

$$\left| \frac{\partial^2 \left[\mathcal{P}(\phi_{i'})\right]}{\partial \alpha_{i',j'} \partial \alpha_{i',j''}} \right| \leq \max\{p, 1-p\} \cdot \frac{(\tau+1)}{\tau^2 \beta^2} \leq \frac{\tau+1}{\tau^2 \beta^2},$$

and the second-order partial derivative of $f_\mathcal{D}(\boldsymbol{\alpha}, \boldsymbol{M})$ can be bounded as following:

$$\left| \frac{\partial^2}{\partial \alpha_{i',j'} \partial \alpha_{i',j''}} \left[f_\mathcal{D}(\boldsymbol{\alpha}, \boldsymbol{M})\right] \right| \leq \frac{(\tau+1)U}{\tau^2 \beta^2}.$$

Finally, we define the $H(\boldsymbol{\alpha})$ as the Hessian matrix of $f_\mathcal{D}(\boldsymbol{\alpha}, \boldsymbol{M})$ for $\boldsymbol{\alpha}$, based on the relationship between $\|H(\boldsymbol{\alpha})\|_2$ and $\|H(\boldsymbol{\alpha})\|_F$:

$$\|H(\boldsymbol{\alpha})\|_2 \leq \|H(\boldsymbol{\alpha})\|_F \leq \sqrt{n(n-1)N^2 \left(\frac{U}{\tau^2 \beta^2}\right)^2 + nN \left(\frac{(\tau+1)U}{\tau^2 \beta^2}\right)^2} \leq \frac{nUN(\tau+1)}{\tau^2 \beta^2}.$$

According to **Lemma 1.2.2** in (Nesterov et al., 2018), $f_\mathcal{D}(\boldsymbol{\alpha}, \boldsymbol{M})$ is lipschitz smooth for $\boldsymbol{\alpha}$ with smooth constant $L_{\boldsymbol{\alpha}} = \frac{nUN(\tau+1)}{\tau^2 \beta^2}$, $\qquad\square$

**Lemma 2** (Smoothness for $\boldsymbol{M}$). *We perform a cropping operation on $\boldsymbol{M} = (m_{d,c})_{D \times C}$ and $|m_{d,c}| \geq \xi > 0$ for $d = 1, ..., D$ and $c = 1, ..., C$, then $f_{\mathcal{D}}(\boldsymbol{\alpha}, \boldsymbol{M})$ is lipschitz smooth for $\boldsymbol{M}$ with smooth constant is $L_{\boldsymbol{M}} = \frac{1}{\xi^2}$.*

*Proof.* The objective function:

$$\mathbb{E}_{\Phi \sim \mathcal{S}(\boldsymbol{\alpha})} [\mathcal{L}(\Phi, \boldsymbol{M}; \mathcal{D})] = \sum_{\phi_1 \sim \mathcal{S}(\boldsymbol{\alpha}_1)} \cdots \sum_{\phi_n \sim \mathcal{S}(\boldsymbol{\alpha}_n)} \left( \mathcal{L}(\Phi, \boldsymbol{M}; \mathcal{D}) \cdot \prod_{i=1}^{n} P(\phi_i) \right).$$

And because we use the cross-entropy function:

$$\mathcal{L}(\Phi, \boldsymbol{M}; \mathcal{D}) = \frac{1}{K} \sum_{(\boldsymbol{x}_k, \boldsymbol{y}_k) \in \mathcal{D}} \left\{ -\boldsymbol{y}_k \cdot [\log(\text{Softmax}(\mathcal{G}(\Phi, \boldsymbol{x}_k)) \cdot \boldsymbol{M})]^\top \right\}. \tag{30}$$

We can compute the Hessian of the objective function for $\boldsymbol{M}$, $\forall d', d'' \in 1, \cdots, D$ and $c', c'' \in 1, \cdots, C$

$$\frac{\partial^2}{\partial m_{d',c'} \partial m_{d'',c''}} \mathbb{E}_{\Phi \sim \mathcal{S}(\boldsymbol{\alpha})} [\mathcal{L}(\Phi, \boldsymbol{M}; \mathcal{D})]$$

$$= \sum_{\phi_1 \sim \mathcal{S}(\boldsymbol{\alpha}_1)} \cdots \sum_{\phi_n \sim \mathcal{S}(\boldsymbol{\alpha}_n)} \left( \frac{\partial^2}{\partial m_{d',c'} \partial m_{d'',c''}} \mathcal{L}(\Phi, \boldsymbol{M}; \mathcal{D}) \cdot \prod_{i=1}^{n} P(\phi_i) \right)$$

$$= \sum_{\phi_1 \sim \mathcal{S}(\boldsymbol{\alpha}_1)} \cdots \sum_{\phi_n \sim \mathcal{S}(\boldsymbol{\alpha}_n)} \left( \frac{1}{K} \sum_{(\boldsymbol{x}_k, \boldsymbol{y}_k) \in \mathcal{D}} \left[ \frac{\partial^2 \left( -\boldsymbol{y}_k \cdot [\log(\text{Softmax}(\mathcal{G}(\Phi, \boldsymbol{x}_k)) \cdot \boldsymbol{M})]^\top \right)}{\partial m_{d',c'} \partial m_{d'',c''}} \right] \cdot \prod_{i=1}^{n} P(\phi_i) \right).$$

We note that $\boldsymbol{y}_k = (y_{k,1}, y_{k,2}, ..., y_{k,C})$ is a one-hot vector, and we abbreviate LLM model's output $\text{Softmax}(\mathcal{G}(\Phi, \boldsymbol{x}_k))$ as $\mathcal{G}_k$, $\mathcal{G}_k = (\mathcal{G}_{k,1}, \mathcal{G}_{k,2}, ..., \mathcal{G}_{k,D})$ is a normalized vector by Softmax function, then we compute the Hessian matrix of the cross-entropy loss function with respect to $\boldsymbol{M}$:

$$\frac{\partial^2 \left( -\boldsymbol{y}_k \cdot [\log(\text{Softmax}(\mathcal{G}(\Phi, \boldsymbol{x}_k)) \cdot \boldsymbol{M})]^\top \right)}{\partial m_{d',c'} \partial m_{d'',c''}} = \begin{cases} \frac{y_{k,c'} \cdot \mathcal{G}_{k,d'} \cdot \mathcal{G}_{k,d''}}{\left( \sum_{d=1}^{D} \mathcal{G}_{k,d} \cdot m_{d,c'} \right)^2}, & \text{if} \quad c' = c''; \\ 0, & \text{if} \quad c' \neq c''. \end{cases}$$

Then, we can get:

$$\frac{\partial^2}{\partial m_{d',c'} \partial m_{d'',c''}} \mathbb{E}_{\Phi \sim \mathcal{S}(\boldsymbol{\alpha})} [\mathcal{L}(\Phi, \boldsymbol{M}; \mathcal{D})]$$

$$= \sum_{\phi_1 \sim \mathcal{S}(\boldsymbol{\alpha}_1)} \cdots \sum_{\phi_n \sim \mathcal{S}(\boldsymbol{\alpha}_n)} \left( \frac{1}{K} \sum_{k=1} \left[ \frac{y_{k,c'} \cdot \mathcal{G}_{k,d'} \cdot \mathcal{G}_{k,d''}}{\left( \sum_{d=1}^{D} \mathcal{G}_{k,d} \cdot m_{d,c'} \right)^2} \right] \cdot \prod_{i=1}^{n} P(\phi_i) \right)$$

$$= \frac{1}{K} \sum_{k=1} \left[ \frac{y_{k,c'} \cdot \mathcal{G}_{k,d'} \cdot \mathcal{G}_{k,d''}}{\left( \sum_{d=1}^{D} \mathcal{G}_{k,d} \cdot m_{d,c'} \right)^2} \right].$$

Without loss of generality, because $\boldsymbol{y}_k$ is a one-hot vector, we assume that:

$$y_{k,c} = \begin{cases} 1, & \text{if} \quad c = c^*; \\ 0, & \text{if} \quad c \neq c^*. \end{cases} \tag{31}$$

So, we can get:

$$\frac{\partial^2}{\partial m_{d',c'} \partial m_{d'',c''}} \mathbb{E}_{\Phi \sim \mathcal{S}(\boldsymbol{\alpha})} [\mathcal{L}(\Phi, \boldsymbol{M}; \mathcal{D})] = \frac{1}{K} \sum_{k=1} \left[ \frac{\mathcal{G}_{k,d'} \cdot \mathcal{G}_{k,d''}}{\left( \sum_{d=1}^{D} \mathcal{G}_{k,d} \cdot m_{d,c^*} \right)^2} \right].$$

Then, with $H(\boldsymbol{M})$ denoting the Hessian matrix of $f_{\mathcal{D}}(\boldsymbol{\alpha}, \boldsymbol{M})$ for $\boldsymbol{M}$, we can obtain an upper bound for $\|H(\boldsymbol{M})\|_F$:

$$
\|H(\boldsymbol{M})\|_F = \sqrt{\sum_{d'=1}^{D} \sum_{d''=1}^{D} \left( \frac{1}{K} \sum_{k=1}^{K} \left[ \frac{\mathcal{G}_{k,d'} \cdot \mathcal{G}_{k,d''}}{\left( \sum_{d=1}^{D} \mathcal{G}_{k,d} \cdot m_{d,c^*} \right)^2} \right] \right)^2}
$$

$$
\overset{(1)}{\leq} \sqrt{\frac{1}{K} \sum_{k=1}^{K} \sum_{d'=1}^{D} \sum_{d''=1}^{D} \left[ \frac{\mathcal{G}_{k,d'} \cdot \mathcal{G}_{k,d''}}{\left( \sum_{d=1}^{D} \mathcal{G}_{k,d} \cdot m_{d,c^*} \right)^2} \right]^2}
$$

$$
= \sqrt{\frac{1}{K} \sum_{k=1}^{K} \frac{\sum_{d'=1}^{D} \left( \mathcal{G}_{k,d'} \right)^2 \cdot \sum_{d''=1}^{D} \left( \mathcal{G}_{k,d''} \right)^2}{\left( \sum_{d=1}^{D} \mathcal{G}_{k,d} \cdot m_{d,c^*} \right)^4_2}}
$$

$$
\overset{(2)}{\leq} \sqrt{\frac{1}{K} \sum_{k=1}^{K} \frac{\left( \sum_{d'=1}^{D} \mathcal{G}_{k,d'} \right) \cdot \left( \sum_{d''=1}^{D} \mathcal{G}_{k,d''} \right)}{\left( \sum_{d=1}^{D} \mathcal{G}_{k,d} \cdot m_{d,c^*} \right)^4}}
$$

$$
\overset{(3)}{=} \sqrt{\frac{1}{K} \sum_{k=1}^{K} \frac{1}{\left( \sum_{d=1}^{D} \mathcal{G}_{k,d} \cdot m_{d,c^*} \right)^4}} \overset{(4)}{\leq} \frac{1}{\xi^2}. \tag{32}
$$

**Note:**

- (1) use inequality: $\left\| \sum_{z=1}^{Z} a_z \right\|^2 \leq Z \sum_{z=1}^{Z} \|a_z\|^2$.

- (2) and (3) is because $\mathcal{G}_k$ is a normalized vector by Softmax function.

- (4) is use $|m_{d,c}| \geq \xi$ for $d = 1, ..., D$ and $c = 1, ..., C$.

Further, based on the relationship between $\|H(\boldsymbol{M})\|_2$ and $\|H(\boldsymbol{M})\|_F$:

$$
\|H(\boldsymbol{M})\|_2 \leq \|H(\boldsymbol{M})\|_F \leq \frac{1}{\xi^2}.
$$

According to **Lemma 1.2.2** in (Nesterov et al., 2018), $f_{\mathcal{D}}(\boldsymbol{\alpha}, \boldsymbol{M})$ is lipschitz smooth for $\boldsymbol{M}$ with smooth constant $L_{\boldsymbol{M}} = \frac{1}{\xi^2}$. $\qquad \square$

**Lemma 3** (Unbiasedness and bounded variance of prompt sampling for $\boldsymbol{\alpha}$). *Let $\alpha_{i,j} \geq \beta > 0$ for $i = 1, ..., n$ and $j = 1, ..., N$, $\tau > 0$ is the temperature parameter, and $\tilde{\sigma}_{\boldsymbol{\alpha}}^2 = \frac{8U^2 N}{\tau^2 \beta^2}$, then the variance-reduced policy gradient of $\boldsymbol{\alpha}_i$ is unbiased and its variance is bounded by $\tilde{\sigma}_{\boldsymbol{\alpha}}^2$:*

$$
\mathbb{E}_{\{\Phi^r \sim \mathcal{S}(\boldsymbol{\alpha})\}_{r=1}^{I_{\boldsymbol{\alpha}}}} \left[ \hat{\nabla}_{\boldsymbol{\alpha}_i} f_k(\boldsymbol{\alpha}, \boldsymbol{M}) \right] = \nabla_{\boldsymbol{\alpha}_i} f_k(\boldsymbol{\alpha}, \boldsymbol{M}); \tag{33}
$$

$$
\mathbb{E}_{\{\Phi^r \sim \mathcal{S}(\boldsymbol{\alpha})\}_{r=1}^{I_{\boldsymbol{\alpha}}}} \left\| \hat{\nabla}_{\boldsymbol{\alpha}_i} f_k(\boldsymbol{\alpha}, \boldsymbol{M}) - \nabla_{\boldsymbol{\alpha}_i} f_k(\boldsymbol{\alpha}, \boldsymbol{M}) \right\|_2^2 \leq \frac{\tilde{\sigma}_{\boldsymbol{\alpha}}^2}{I_{\boldsymbol{\alpha}}^2}. \tag{34}
$$

*Proof.* First we proof the variance-reduced policy gradient for $\boldsymbol{\alpha}_i$ is unbiased, according to the independence of each sampling for $\Phi^r$, $r = 1, ..., I_{\boldsymbol{\alpha}}$:

$$
\mathbb{E}_{\{\Phi^r \sim \mathcal{S}(\boldsymbol{\alpha})\}_{r=1}^{I_{\boldsymbol{\alpha}}}} \left[ \hat{\nabla}_{\boldsymbol{\alpha}_i} f_k(\boldsymbol{\alpha}, \boldsymbol{M}) \right]
$$

$$
= \mathbb{E}_{\{\Phi^r\}_{r=1}^{I_{\boldsymbol{\alpha}}}} \left[ \frac{1}{I_{\boldsymbol{\alpha}} - 1} \sum_{r=1}^{I_{\boldsymbol{\alpha}}} \left( \mathcal{L} \left( \Phi^r, \boldsymbol{M}; \boldsymbol{d}_k \right) - \frac{1}{I_{\boldsymbol{\alpha}}} \sum_{\gamma=1}^{I_{\boldsymbol{\alpha}}} \mathcal{L} \left( \Phi^\gamma, \boldsymbol{M}; \boldsymbol{d}_k \right) \right) \nabla_{\boldsymbol{\alpha}_i} \log \mathcal{P}(\phi_i^r) \right]
$$

$$
= \mathbb{E}_{\{\Phi^r\}_{r=1}^{I_{\boldsymbol{\alpha}}}} \left[ \frac{1}{I_{\boldsymbol{\alpha}} - 1} \sum_{r=1}^{I_{\boldsymbol{\alpha}}} \left( \frac{I_{\boldsymbol{\alpha}} - 1}{I_{\boldsymbol{\alpha}}} \cdot \mathcal{L}\left(\Phi^r, \boldsymbol{M}; \boldsymbol{d}_k\right) - \frac{1}{I_{\boldsymbol{\alpha}}} \sum_{\substack{\gamma = 1 \\ \gamma \neq r}}^{I_{\boldsymbol{\alpha}}} \mathcal{L}\left(\Phi^\gamma, \boldsymbol{M}; \boldsymbol{d}_k\right) \right) \nabla_{\boldsymbol{\alpha}_i} \log \mathcal{P}(\phi_i^r) \right]
$$

$$
= \frac{1}{I_{\boldsymbol{\alpha}}} \sum_{r=1}^{I_{\boldsymbol{\alpha}}} \mathbb{E}_{\Phi^r} \left[ \mathcal{L}\left(\Phi^r, \boldsymbol{M}; \boldsymbol{d}_k\right) \cdot \nabla_{\boldsymbol{\alpha}_i} \log \mathcal{P}(\phi_i^r) \right]
$$

$$
- \frac{1}{I_{\boldsymbol{\alpha}}} \sum_{r=1}^{I_{\boldsymbol{\alpha}}} \left[ \left( \frac{1}{I_{\boldsymbol{\alpha}} - 1} \sum_{\substack{\gamma = 1 \\ \gamma \neq r}}^{I_{\boldsymbol{\alpha}}} \mathbb{E}_{\Phi^\gamma} \mathcal{L}\left(\Phi^\gamma, \boldsymbol{M}; \boldsymbol{d}_k\right) \right) \mathbb{E}_{\Phi^r} \nabla_{\boldsymbol{\alpha}_i} \log \mathcal{P}(\phi_i^r) \right]
$$

$$
= \nabla_{\boldsymbol{\alpha}_i} f_k(\boldsymbol{\alpha}, \boldsymbol{M}) - \frac{1}{I_{\boldsymbol{\alpha}}} \sum_{r=1}^{I_{\boldsymbol{\alpha}}} \left[ \left( \frac{1}{I_{\boldsymbol{\alpha}} - 1} \sum_{\substack{\gamma = 1 \\ \gamma \neq r}}^{I_{\boldsymbol{\alpha}}} \mathbb{E}_{\Phi^\gamma} \left[ \mathcal{L}\left(\Phi^\gamma, \boldsymbol{M}; \boldsymbol{d}_k\right) \right] \right) \mathbb{E}_{\Phi^r} \left[ \nabla_{\boldsymbol{\alpha}_i} \log \mathcal{P}(\phi_i^r) \right] \right] .
$$

$$(35)$$

Then, for the second item of (35):

$$
\frac{1}{I_{\boldsymbol{\alpha}}} \sum_{r=1}^{I_{\boldsymbol{\alpha}}} \left[ \left( \frac{1}{I_{\boldsymbol{\alpha}} - 1} \sum_{\substack{\gamma = 1 \\ \gamma \neq r}}^{I_{\boldsymbol{\alpha}}} \mathbb{E}_{\Phi^\gamma} \left[ \mathcal{L}\left(\Phi^\gamma, \boldsymbol{M}; \boldsymbol{d}_k\right) \right] \right) \cdot \mathbb{E}_{\Phi^r} \left[ \nabla_{\boldsymbol{\alpha}_i} \log \mathcal{P}(\phi_i^r) \right] \right]
$$

$$
\overset{(1)}{=} \frac{1}{I_{\boldsymbol{\alpha}}} \sum_{r=1}^{I_{\boldsymbol{\alpha}}} \left[ \left( \frac{1}{I_{\boldsymbol{\alpha}} - 1} \sum_{\substack{\gamma = 1 \\ \gamma \neq r}}^{I_{\boldsymbol{\alpha}}} \mathbb{E}_{\Phi^\gamma} \left[ \mathcal{L}\left(\Phi^\gamma, \boldsymbol{M}; \boldsymbol{d}_k\right) \right] \right) \cdot \sum_{\phi_i^r \sim \boldsymbol{p}_i} \left[ \nabla_{\boldsymbol{\alpha}_i} \mathcal{P}(\phi_i^r) \right] \right]
$$

$$
\overset{(2)}{=} \frac{1}{I_{\boldsymbol{\alpha}}} \sum_{r=1}^{I_{\boldsymbol{\alpha}}} \left[ \left( \frac{1}{I_{\boldsymbol{\alpha}} - 1} \sum_{\substack{\gamma = 1 \\ \gamma \neq r}}^{I_{\boldsymbol{\alpha}}} \mathbb{E}_{\Phi^\gamma} \left[ \mathcal{L}\left(\Phi^\gamma, \boldsymbol{M}; \boldsymbol{d}_k\right) \right] \right) \cdot \nabla_{\boldsymbol{\alpha}_i} \left( \sum_{\phi_i^r \sim \boldsymbol{p}_i} \mathcal{P}(\phi_i^r) \right) \right]
$$

$$
\overset{(3)}{=} \frac{1}{I_{\boldsymbol\alpha}} \sum_{r=1}^{I_{\boldsymbol\alpha}} \left[ \left( \frac{1}{I_{\boldsymbol\alpha}-1} \sum_{\substack{\gamma=1 \\ \gamma \neq r}}^{I_{\boldsymbol\alpha}} \mathbb{E}_{\Phi^\gamma} \left[ \mathcal{L}\left(\Phi^\gamma, \boldsymbol{M}; \boldsymbol{d}_k\right) \right] \right) \cdot \nabla_{\boldsymbol\alpha_i}(1) \right]
$$

$$
= 0. \tag{36}
$$

**Note:**

- (1) is because the sampling process with respect to $\Phi^r$ is discrete.

- (2) is because the number of prompt token $n$ is not infinite and the $\mathcal{S}$ function is derivable with respect to $\boldsymbol\alpha_i$.

- (3) uses the normalisation property of $\mathcal{S}$ function.

We substitute (36) into (35) to obtain:

$$
\mathbb{E}_{\{\Phi^r \sim \mathcal{S}(\alpha)\}_{r=1}^{I_{\boldsymbol\alpha}}} \left[ \hat{\nabla}_{\boldsymbol\alpha_i} f_k(\boldsymbol\alpha, \boldsymbol{M}) \right] = \nabla_{\boldsymbol\alpha_i} f_k(\boldsymbol\alpha, \boldsymbol{M}).
$$

Then, we proof the bounded variance of variance-reduced policy gradient for $\boldsymbol\alpha_i$:

$$
\mathbb{E}_{\{\Phi^r \sim \mathcal{S}(\alpha)\}_{r=1}^{I_{\boldsymbol\alpha}}} \left\| \hat{\nabla}_{\boldsymbol\alpha_i} f_k(\boldsymbol\alpha, \boldsymbol{M}) - \nabla_{\boldsymbol\alpha_i} f_k(\boldsymbol\alpha, \boldsymbol{M}) \right\|_2^2
$$

$$
= \mathbb{E}_{\{\Phi^r\}_{r=1}^{I_{\boldsymbol\alpha}}} \left\| \frac{1}{I_{\boldsymbol\alpha}} \sum_{r=1}^{I_{\boldsymbol\alpha}} \left\{ \frac{1}{I_{\boldsymbol\alpha}-1} \sum_{\substack{\gamma=1 \\ \gamma \neq r}}^{I_{\boldsymbol\alpha}} \left[ \mathcal{L}\left(\Phi^r, \boldsymbol{M}; \boldsymbol{d}_k\right) - \mathcal{L}\left(\Phi^\gamma, \boldsymbol{M}; \boldsymbol{d}_k\right) \right] \nabla_{\boldsymbol\alpha_i} \log \mathcal{P}(\phi_i^r) - \nabla_{\boldsymbol\alpha_i} f_k(\boldsymbol\alpha, \boldsymbol{M}) \right\} \right\|_2^2
$$

$$
\overset{(1)}{=} \frac{1}{I_{\boldsymbol\alpha}^2} \sum_{r=1}^{I_{\boldsymbol\alpha}} \mathbb{E}_{\Phi^r} \left\| \frac{1}{I_{\boldsymbol\alpha}-1} \sum_{\substack{\gamma=1 \\ \gamma \neq r}}^{I_{\boldsymbol\alpha}} \left[ \mathcal{L}\left(\Phi^r, \boldsymbol{M}; \boldsymbol{d}_k\right) - \mathcal{L}\left(\Phi^\gamma, \boldsymbol{M}; \boldsymbol{d}_k\right) \right] \nabla_{\boldsymbol\alpha_i} \log \mathcal{P}(\phi_i^r) - \nabla_{\boldsymbol\alpha_i} f_k(\boldsymbol\alpha, \boldsymbol{M}) \right\|_2^2
$$

$$
\overset{(2)}{\leq} \frac{1}{I_{\boldsymbol\alpha}^2(I_{\boldsymbol\alpha}-1)^2} \sum_{r=1}^{I_{\boldsymbol\alpha}} \mathbb{E}_{\Phi^r} \left\| \sum_{\substack{\gamma=1 \\ \gamma \neq r}}^{I_{\boldsymbol\alpha}} \left[ \mathcal{L}\left(\Phi^r, \boldsymbol{M}; \boldsymbol{d}_k\right) - \mathcal{L}\left(\Phi^\gamma, \boldsymbol{M}; \boldsymbol{d}_k\right) \right] \nabla_{\boldsymbol\alpha_i} \log \mathcal{P}(\phi_i^r) \right\|_2^2
$$

$$
\overset{(3)}{\leq} \frac{4U^2}{I_{\boldsymbol\alpha}^2(I_{\boldsymbol\alpha}-1)^2} \sum_{r=1}^{I_{\boldsymbol\alpha}} \mathbb{E}_{\Phi^r} \left\| \sum_{\substack{\gamma=1 \\ \gamma \neq r}}^{I_{\boldsymbol\alpha}} \nabla_{\boldsymbol\alpha_i} \log \mathcal{P}(\phi_i^r) \right\|_2^2 \overset{(4)}{\leq} \frac{4U^2 N}{I_{\boldsymbol\alpha}(I_{\boldsymbol\alpha}-1)\tau^2\beta^2} \overset{(5)}{\leq} \frac{8U^2 N}{I_{\boldsymbol\alpha}^2 \tau^2 \beta^2}.
$$

**Note:**

- (1) is because the independence of each sampling for $\Phi$, and:

$$\mathbb{E}_{\{\Phi^r\}_{r=1}^{I_{\boldsymbol{\alpha}}}} \left\{ \frac{1}{I_{\boldsymbol{\alpha}}-1} \sum_{\substack{\gamma=1 \\ \gamma \neq r}}^{I_{\boldsymbol{\alpha}}} \left[\mathcal{L}\left(\Phi^r, \boldsymbol{M}; \boldsymbol{d}_k\right) - \mathcal{L}\left(\Phi^\gamma, \boldsymbol{M}; \boldsymbol{d}_k\right)\right] \nabla_{\boldsymbol{\alpha}_i} \log \mathcal{P}(\phi_i^r) \right\}$$

$$= \frac{1}{I_{\boldsymbol{\alpha}}-1} \sum_{\substack{\gamma=1 \\ \gamma \neq r}}^{I_{\boldsymbol{\alpha}}} \mathbb{E}_{\{\Phi^r\}_{r=1}^{I_{\boldsymbol{\alpha}}}} \left[\mathcal{L}\left(\Phi^r, \boldsymbol{M}; \boldsymbol{d}_k\right) - \mathcal{L}\left(\Phi^\gamma, \boldsymbol{M}; \boldsymbol{d}_k\right) \nabla_{\boldsymbol{\alpha}_i} \log \mathcal{P}(\phi_i^r)\right]$$

$$= \nabla_{\boldsymbol{\alpha}_i} f_k(\boldsymbol{\alpha}, \boldsymbol{M}). \tag{37}$$

- (2) uses inequality $\mathbb{E}\|a - \mathbb{E}a\|_2^2 \leq \mathbb{E}\|a\|_2^2$.

- (3) uses **Assumption 3**.

- (4) uses $\alpha_{i,j}^r \geq \beta > 0$ and (6):

$$\nabla_{\boldsymbol{\alpha}_i} \log \mathcal{P}(\phi_i^r) \leq \sqrt{N \cdot \max\left\{\left|\frac{1-p_{i,j_i}^r}{\tau \alpha_{i,j_i}^r}\right|, \left|-\frac{p_{i,j}^r}{\tau \alpha_{i,j}^r}\right|\right\}^2} \leq \sqrt{\frac{N}{\tau^2 \beta^2}}.$$

- (5) is because when $I \geq 2$:

$$\frac{1}{I_{\boldsymbol{\alpha}}(I_{\boldsymbol{\alpha}}-1)} \leq \frac{2}{I_{\boldsymbol{\alpha}}^2}.$$

Finally, let $\tilde{\sigma}_{\boldsymbol{\alpha}}^2 = \frac{8U^2 N}{\tau^2 \beta^2}$, and proof is completed. $\qquad\square$

**Lemma 4** (Unbiasedness and bounded variance of prompt sampling for $\boldsymbol{M}$). *We perform a cropping operation on $\boldsymbol{M} = (m_{d,c})_{D \times C}$ and $|m_{d,c}| \geq \xi$ for $d = 1, ..., D$ and $c = 1, ..., C$, $\tau > 0$ is the temperature parameter, and $\tilde{\sigma}_{\boldsymbol{M}}^2 = \frac{4}{\xi^2}$, then the gradient with prompt sampling of $\boldsymbol{M}$ is unbiased and its variance is bounded by :*

$$\mathbb{E}_{\{\Phi^s \sim \mathcal{S}(\boldsymbol{\alpha})\}_{s=1}^{I_M}} \left[\tilde{\nabla}_{\boldsymbol{M}} f_k(\boldsymbol{\alpha}, \boldsymbol{M})\right] = \nabla_{\boldsymbol{M}} f_k(\boldsymbol{\alpha}, \boldsymbol{M}); \tag{38}$$

$$\mathbb{E}_{\{\Phi^s \sim \mathcal{S}(\boldsymbol{\alpha})\}_{s=1}^{I_M}} \left\|\tilde{\nabla}_{\boldsymbol{M}} f_k(\boldsymbol{\alpha}, \boldsymbol{M}) - \nabla_{\boldsymbol{M}} f_k(\boldsymbol{\alpha}, \boldsymbol{M})\right\|_2^2 \leq \frac{\tilde{\sigma}_{\boldsymbol{M}}^2}{I_M}. \tag{39}$$

*Proof.* First we proof the gradient with prompt sampling for $\boldsymbol{M}$ is unbiased:

$$\mathbb{E}_{\{\Phi^s \sim \mathcal{S}(\boldsymbol{\alpha})\}_{s=1}^{I_M}} \left[\tilde{\nabla}_{\boldsymbol{M}} f_k(\boldsymbol{\alpha}, \boldsymbol{M})\right]$$

$$= \mathbb{E}_{\{\Phi^s\}_{s=1}^{I_M}} \left(\frac{1}{I_M} \sum_{s=1}^{I_M} \nabla_{\boldsymbol{M}} \mathcal{L}\left(\Phi^s, \boldsymbol{M}; \boldsymbol{d}_k\right)\right)$$

$$\stackrel{(1)}{=} \frac{1}{I_M} \sum_{s=1}^{I_M} \mathbb{E}_{\Phi^s} \left[\nabla_{\boldsymbol{M}} \mathcal{L}\left(\Phi^s, \boldsymbol{M}; \boldsymbol{d}_k\right)\right]$$

$$\stackrel{(2)}{=} \frac{1}{I_M} \sum_{s=1}^{I_M} \nabla_{\boldsymbol{M}} \mathbb{E}_{\Phi^s} \left[\mathcal{L}\left(\Phi^s, \boldsymbol{M}; \boldsymbol{d}_k\right)\right]$$

$$= \nabla_{\boldsymbol{M}} f_k(\boldsymbol{\alpha}, \boldsymbol{M}).$$

where (1) use the independence of each sampling for $\Phi^s$, $s = 1, ..., I_M$; (2) is because $\mathbb{E}_\Phi$ can be expanded as the sum of the products of a finite number of probabilities and random variables (3) and $\mathcal{L}(\Phi, M; d_k)$ is differentiable with respect to $M$.

Then, we proof the bounded variance of the gradient with prompt sampling for $M$:

$$\mathbb{E}_{\{\Phi^s \sim \mathcal{S}(\alpha)\}_{s=1}^{I_M}} \left\| \tilde{\nabla}_M f_k(\alpha, M) - \nabla_M f_k(\alpha, M) \right\|_2^2$$

$$= \mathbb{E}_{\{\Phi^s\}_{s=1}^{I_M}} \left\| \frac{1}{I_M} \sum_{s=1}^{I_M} \nabla_M \mathcal{L}(\Phi^s, M; d_k) - \nabla_M \mathbb{E}_{\Phi \sim \mathcal{S}(\alpha)} [\mathcal{L}(\Phi, M; d_k)] \right\|_2^2$$

$$\overset{(1)}{=} \mathbb{E}_{\{\Phi^s\}_{s=1}^{I_M}} \left\| \frac{1}{I_M} \sum_{s=1}^{I_M} [\nabla_M \mathcal{L}(\mathcal{L}(\Phi^s, M; d_k)) - \mathbb{E}_{\Phi \sim \mathcal{S}(\alpha)} [\nabla_M \mathcal{L}(\Phi, M; d_k)]] \right\|_2^2$$

$$\overset{(2)}{=} \frac{1}{I_M^2} \sum_{s=1}^{I_M} \mathbb{E}_{\Phi^s} \left\| \nabla_M \mathcal{L}(\Phi^s, M; d_k) - \mathbb{E}_{\Phi \sim \mathcal{S}(\alpha)} [\nabla_M \mathcal{L}(\Phi, M; d_k)] \right\|_2^2.$$

where (1) is because $\mathbb{E}_\Phi$ can be expanded as the sum of the products of a finite number of probabilities and random variables (3) and $\mathcal{L}(\Phi, M; d_k)$ is differentiable with respect to $M$; (2) is because the sampling of $\Phi^s$ is independent.

We note that $y_k = (y_{k,1}, y_{k,2}, ..., y_{k,C})$ is a one-hot vector, and we abbreviate LLM model's output Softmax $(\mathcal{G}(\Phi, x_k))$ as $\mathcal{G}_k$, $\mathcal{G}_k = (\mathcal{G}_{k,1}, \mathcal{G}_{k,2}, ..., \mathcal{G}_{k,D})$ is a normalized vector by Softmax function, since the $\mathcal{L}$ function is the cross entropy function, we calculate its derivative for $M$ as follows:

$$\nabla_M \mathcal{L}(\Phi, M; d_k) = \frac{\left( -y_k \cdot [\log(\text{Softmax}(\mathcal{G}(\Phi, x_k)) \cdot M)]^\top \right)}{\partial m_{d',c'}} = \left( -\frac{y_{k,c'} \cdot \mathcal{G}_{k,d'}}{\sum_{d=1}^D \mathcal{G}_{k,d} \cdot m_{d,c'}} \right)_{D \times C}.$$

where $d' = 1, ...D$ and $c' = 1, ..., C$.

Then, we can get the upper bound of the $\ell_2$-norm for $\nabla_M \mathcal{L}(\mathcal{G}(\Phi, x_k) \cdot M, y_k)$ as following:

$$\|\nabla_M \mathcal{L}(\Phi, M; d_k)\|_2^2 = \left\| \left( -\frac{y_{k,c'} \cdot \mathcal{G}_{k,d'}}{\sum_{d=1}^D \mathcal{G}_{k,d} \cdot m_{d,c'}} \right)_{D \times C} \right\|_2^2.$$

Without loss of generality, because $y_k$ is a one-hot vector, we assume that:

$$y_{k,c} = \begin{cases} 1, & \text{if} \quad c = c^*; \\ 0, & \text{if} \quad c \neq c^*. \end{cases}$$

Then:

$$\|\nabla_M \mathcal{L}(\Phi, M; d_k)\|_2^2$$

$$= \left\| \left( -\frac{\mathcal{G}_{k,d'}}{\sum_{d=1}^D \mathcal{G}_{k,d} \cdot m_{d,c^*}} \right)_{D \times 1} \right\|_2^2$$

$$\overset{(1)}{\leq} \frac{\sum_{d'=1}^D \mathcal{G}_{k,d'}}{\left( \sum_{d=1}^D \mathcal{G}_{k,d} \cdot m_{d,c^*} \right)^2} \overset{(2)}{\leq} \frac{1}{\xi^2}. \tag{40}$$

where (1) is because $0 < \mathcal{G}_{k,d'} < 1$; (2) is use $|m_{d,c}| \geq \xi > 0$ for $d = 1, ..., D$ and $c = 1, ..., C$, $\mathcal{G}_k$ is a normalized vector by Softmax function.

Finally:

$$\mathbb{E}_{\{\Phi^s \sim \mathcal{S}(\alpha)\}_{s=1}^{I_M}} \left\| \tilde{\nabla}_M f_k(\alpha, M) - \mathbb{E}_{\{\Phi^s \sim \mathcal{S}(\alpha)\}_{s=1}^{I_M}} \left[ \tilde{\nabla}_M f_k(\alpha, M) \right] \right\|_2^2$$

$$\leq \frac{2}{I_M^2} \sum_{s=1}^{I_M} \mathbb{E}_{\Phi^s} \|\nabla_M \mathcal{L}(\Phi^s, M; d_k)\|_2^2 + \frac{2}{I_M^2} \sum_{s=1}^{I_M} \mathbb{E}_{\Phi^s} \left\| \mathbb{E}_{\Phi \sim \mathcal{S}(\alpha)} [\nabla_M \mathcal{L}(\Phi, M; d_k)] \right\|_2^2$$

$$\leq \frac{2}{I_M^2} \frac{I_M}{\xi^2} + \frac{2}{I_M^2} \frac{I_M}{\xi^2}$$

$$= \frac{4}{I_M \xi^2}.$$

Finally, let $\tilde{\sigma}_M^2 = \frac{4}{\xi^2}$, and proof is completed. $\qquad \square$

## A.5 CONVERGENCE OF LEAP

**Theorem 1** (Convergence of LEAP). *Suppose **Assumption 1**, **2** and **3** hold, for iteration $t = 0, ..., T-1$, set $\alpha_{i,j} \geq \beta > 0$ and $|m_{d,c}| \geq \xi > 0$, $\tau > 0$ is the temperature parameter, $f_{\mathcal{D}}(\alpha, M)$ is smooth for $\alpha$ with smooth constant $L_{\alpha} = \frac{nUN(\tau+1)}{\tau^2\beta^2}$ and lipschitz smooth for $M$ with smooth constant is $L_M = \frac{1}{\xi^2}$, $\sigma_{\alpha}^2$ and $\sigma_M^2$ are the variance of the stochastic gradient for $\alpha$ and $M$, $\tilde{\sigma}_{\alpha}^2 = \frac{8U^2N}{\tau^2\beta^2}$ and $\tilde{\sigma}_M^2 = \frac{4}{\xi^2}$ are the variance of prompt sampling for $\alpha$ and $M$. We define $\eta_{min} = \min\{\eta_{\alpha}, \eta_M\}$ and $\eta_{max} = \max\{\eta_{\alpha}, \eta_M\}$, and run **Algorithm 1** with $0 < \eta_{\alpha} < \frac{1}{L_{\alpha}}$, $0 < \eta_M < \frac{1}{L_M}$ and $q_{\eta} = \frac{\eta_{max}}{\eta_{min}} < \infty$, then the LEAP's full gradient satisfies the following inequality:*

$$\frac{1}{T}\sum_{t=0}^{T-1}\left(\|\nabla_{\alpha}f_{\mathcal{D}}(\alpha_t, M_t)\|_2^2 + \|g_{\mathcal{D}}(\alpha_{t+1}, M_t)\|_2^2\right)$$

$$\leq \frac{2\triangle}{T\eta_{min}} + \frac{2nq_{\eta}\tilde{\sigma}_{\alpha}^2}{I_{\alpha}^2} + \frac{4q_{\eta}\tilde{\sigma}_M^2}{I_M} + \frac{2nq_{\eta}\sigma_{\alpha}^2 + 4q_{\eta}\sigma_M^2}{B}. \tag{41}$$

*Proof.* According to the lipschitz smoothness of $\alpha$ in **Lemma 1**:

$$f_{\mathcal{D}}(\alpha_{t+1}, M_t) - f_{\mathcal{D}}(\alpha_t, M_t) \leq \langle\nabla_{\alpha}f_{\mathcal{D}}(\alpha_t, M_t), \alpha_{t+1} - \alpha_t\rangle + \frac{L_{\alpha}}{2}\|\alpha_{t+1} - \alpha_t\|_2^2. \tag{42}$$

According to the lipschitz smoothness of $M$ in **Lemma 2**:

$$f_{\mathcal{D}}(\alpha_{t+1}, M_{t+1}) - f_{\mathcal{D}}(\alpha_{t+1}, M_t) \leq \langle\nabla_M f_{\mathcal{D}}(\alpha_{t+1}, M_t), M_{t+1} - M_t\rangle + \frac{L_M}{2}\|M_{t+1} - M_t\|_2^2. \tag{43}$$

Adding (42) and (43) gives:

$$f_{\mathcal{D}}(\alpha_{t+1}, M_{t+1}) - f_{\mathcal{D}}(\alpha_t, M_t)$$

$$\leq \langle\nabla_{\alpha}f_{\mathcal{D}}(\alpha_t, M_t), \alpha_{t+1} - \alpha_t\rangle + \frac{L_{\alpha}}{2}\|\alpha_{t+1} - \alpha_t\|_2^2$$

$$+ \langle\nabla_M f_{\mathcal{D}}(\alpha_{t+1}, M_t), M_{t+1} - M_t\rangle + \frac{L_M}{2}\|M_{t+1} - M_t\|_2^2$$

$$\leq \sum_{i=1}^{n}\underbrace{\left[\langle\nabla_{\alpha_i}f_{\mathcal{D}}(\alpha_t, M_t), \alpha_{i,t+1} - \alpha_{i,t}\rangle + \frac{L_{\alpha}}{2}\|\alpha_{i,t+1} - \alpha_{i,t}\|_2^2\right]}_{a)}$$

$$+ \underbrace{\langle\nabla_M f_{\mathcal{D}}(\alpha_{t+1}, M_t), M_{t+1} - M_t\rangle + \frac{L_M}{2}\|M_{t+1} - M_t\|_2^2}_{b)}. \tag{44}$$

For a), we let $\eta_{\alpha} < \frac{1}{L_{\alpha}}$, substitute $\alpha_{i,t+1} = \alpha_{i,t} - \eta_{\alpha} \cdot \hat{\nabla}_{\alpha_i}f_{\mathcal{B}}(\alpha_t, M_t)$, take expectations $\mathbb{E}_{\mathcal{B}}$ and $\mathbb{E}_{\{\Phi^r \sim \mathcal{S}(\alpha)\}_{r=1}^{I_{\alpha}}}$ on both sides, and abbreviate $\mathbb{E}_{\{\Phi^r \sim \mathcal{S}(\alpha)\}_{r=1}^{I_{\alpha}}}$ as $\mathbb{E}_{\{\Phi^r\}_{r=1}^{I_{\alpha}}}$:

$$\mathbb{E}_{\mathcal{B}}\mathbb{E}_{\{\Phi^r\}_{r=1}^{I_{\alpha}}}\left[\langle\nabla_{\alpha_i}f_{\mathcal{D}}(\alpha_t, M_t), \alpha_{i,t+1} - \alpha_{i,t}\rangle + \frac{L_{\alpha}}{2}\|\alpha_{i,t+1} - \alpha_{i,t}\|_2^2\right]$$

$$= \mathbb{E}_{\mathcal{B}}\mathbb{E}_{\{\Phi^r\}_{r=1}^{I_{\alpha}}}\left[\left\langle\nabla_{\alpha_i}f_{\mathcal{D}}(\alpha_t, M_t), -\eta_{\alpha} \cdot \hat{\nabla}_{\alpha_i}f_{\mathcal{B}}(\alpha_t, M_t)\right\rangle + \frac{L_{\alpha}\eta_{\alpha}^2}{2}\left\|\hat{\nabla}_{\alpha_i}f_{\mathcal{B}}(\alpha_t, M_t)\right\|_2^2\right]$$

$$\overset{(1)}{=} \langle\nabla_{\alpha_i}f_{\mathcal{D}}(\alpha_t, M_t), -\eta_{\alpha} \cdot \nabla_{\alpha_i}f_{\mathcal{D}}(\alpha_t, M_t)\rangle + \frac{L_{\alpha}\eta_{\alpha}^2}{2}\mathbb{E}_{\mathcal{B}}\mathbb{E}_{\{\Phi^r\}_{r=1}^{I_{\alpha}}}\left\|\hat{\nabla}_{\alpha_i}f_{\mathcal{B}}(\alpha_t, M_t)\right\|_2^2$$

$$\overset{(2)}{=} -\eta_{\alpha}\|\nabla_{\alpha_i}f_{\mathcal{D}}(\alpha_t, M_t)\|_2^2 + \frac{L_{\alpha}\eta_{\alpha}^2}{2}\left\|\mathbb{E}_{\mathcal{B}}\mathbb{E}_{\{\Phi^r\}_{r=1}^{I_{\alpha}}}\hat{\nabla}_{\alpha_i}f_{\mathcal{B}}(\alpha_t, M_t)\right\|_2^2$$

$$+ \frac{L_{\alpha}\eta_{\alpha}^2}{2}\mathbb{E}_{\mathcal{B}}\mathbb{E}_{\{\Phi^r\}_{r=1}^{I_{\alpha}}}\left\|\hat{\nabla}_{\alpha_i}f_{\mathcal{B}}(\alpha_t, M_t) - \mathbb{E}_{\mathcal{B}}\mathbb{E}_{\{\Phi^r\}_{r=1}^{I_{\alpha}}}\hat{\nabla}_{\alpha_i}f_{\mathcal{B}}(\alpha_t, M_t)\right\|_2^2$$

$$\overset{(3)}{\leq} -\eta_{\boldsymbol{\alpha}} \left\| \nabla_{\boldsymbol{\alpha}_i} f_{\mathcal{D}}(\boldsymbol{\alpha}_t, \boldsymbol{M}_t) \right\|_2^2 + \frac{L_{\boldsymbol{\alpha}} \eta_{\boldsymbol{\alpha}}^2}{2} \left\| \nabla_{\boldsymbol{\alpha}_i} f_{\mathcal{D}}(\boldsymbol{\alpha}_t, \boldsymbol{M}_t) \right\|_2^2$$

$$+ L_{\boldsymbol{\alpha}} \eta_{\boldsymbol{\alpha}}^2 \mathbb{E}_{\mathcal{B}} \mathbb{E}_{\{\Phi^r\}_{r=1}^{I_{\boldsymbol{\alpha}}}} \left\| \hat{\nabla}_{\boldsymbol{\alpha}_i} f_{\mathcal{B}}(\boldsymbol{\alpha}_t, \boldsymbol{M}_t) - \mathbb{E}_{\{\Phi^r\}_{r=1}^{I_{\boldsymbol{\alpha}}}} \hat{\nabla}_{\boldsymbol{\alpha}_i} f_{\mathcal{B}}(\boldsymbol{\alpha}_t, \boldsymbol{M}_t) \right\|_2^2$$

$$+ L_{\boldsymbol{\alpha}} \eta_{\boldsymbol{\alpha}}^2 \mathbb{E}_{\{\Phi^r\}_{r=1}^{I_{\boldsymbol{\alpha}}}} \mathbb{E}_{\mathcal{B}} \left\| \mathbb{E}_{\{\Phi^r\}_{r=1}^{I_{\boldsymbol{\alpha}}}} \hat{\nabla}_{\boldsymbol{\alpha}_i} f_{\mathcal{B}}(\boldsymbol{\alpha}_t, \boldsymbol{M}_t) - \mathbb{E}_{\mathcal{B}} \mathbb{E}_{\{\Phi^r\}_{r=1}^{I_{\boldsymbol{\alpha}}}} \hat{\nabla}_{\boldsymbol{\alpha}_i} f_{\mathcal{B}}(\boldsymbol{\alpha}_t, \boldsymbol{M}_t) \right\|_2^2$$

$$\overset{(4)}{\leq} \left( \frac{L_{\boldsymbol{\alpha}} \eta_{\boldsymbol{\alpha}}^2}{2} - \eta_{\boldsymbol{\alpha}} \right) \left\| \nabla_{\boldsymbol{\alpha}_i} f_{\mathcal{D}}(\boldsymbol{\alpha}_t, \boldsymbol{M}_t) \right\|_2^2 + L_{\boldsymbol{\alpha}} \eta_{\boldsymbol{\alpha}}^2 \mathbb{E}_{\mathcal{B}} \left[ \frac{\tilde{\sigma}_{\boldsymbol{\alpha}}^2}{I_{\boldsymbol{\alpha}}^2} \right] + L_{\boldsymbol{\alpha}} \eta_{\boldsymbol{\alpha}}^2 \mathbb{E}_{\{\Phi^r\}_{r=1}^{I_{\boldsymbol{\alpha}}}} \left[ \frac{\sigma_{\boldsymbol{\alpha}}^2}{B} \right]$$

$$= \left( \frac{L_{\boldsymbol{\alpha}} \eta_{\boldsymbol{\alpha}}^2}{2} - \eta_{\boldsymbol{\alpha}} \right) \left\| \nabla_{\boldsymbol{\alpha}_i} f_{\mathcal{D}}(\boldsymbol{\alpha}_t, \boldsymbol{M}_t) \right\|_2^2 + \frac{L_{\boldsymbol{\alpha}} \eta_{\boldsymbol{\alpha}}^2 \tilde{\sigma}_{\boldsymbol{\alpha}}^2}{I_{\boldsymbol{\alpha}}^2} + \frac{L_{\boldsymbol{\alpha}} \eta_{\boldsymbol{\alpha}}^2 \sigma_{\boldsymbol{\alpha}}^2}{B}$$

$$\overset{(5)}{\leq} -\frac{\eta_{\boldsymbol{\alpha}}}{2} \left\| \nabla_{\boldsymbol{\alpha}_i} f_{\mathcal{D}}(\boldsymbol{\alpha}_t, \boldsymbol{M}_t) \right\|_2^2 + \frac{L_{\boldsymbol{\alpha}} \eta_{\boldsymbol{\alpha}}^2 \tilde{\sigma}_{\boldsymbol{\alpha}}^2}{I_{\boldsymbol{\alpha}}^2} + \frac{L_{\boldsymbol{\alpha}} \eta_{\boldsymbol{\alpha}}^2 \sigma_{\boldsymbol{\alpha}}^2}{B}. \tag{45}$$

**Note:**

- (1) use the unbiasedness of stochastic gradient and policy gradient for $\boldsymbol{\alpha}$ in **Assumption 1** and **Lemma 3**.

- (2) use the equality: $\mathbb{E} \left\| a - \mathbb{E}[a] \right\|_2^2 = \mathbb{E} \left\| a \right\|_2^2 - \left\| \mathbb{E}[a] \right\|_2^2$.

- (3) use the inequality: $\left\| a + b \right\|_2^2 \leq 2 \left\| a \right\|_2^2 + 2 \left\| b \right\|_2^2$.

- (4) use the bounded variance of stochastic gradients and gradient with prompt sampling for $M$ in **Assumption 1** and **Lemma 3**.

- (5) use $\eta_{\boldsymbol{\alpha}} < \frac{1}{L_{\boldsymbol{\alpha}}}$.

For b), we substitute $\boldsymbol{M}_{t+1} = \boldsymbol{M}_t - \eta_M \tilde{g}_{\mathcal{B}}(\boldsymbol{\alpha}_{t+1}, \boldsymbol{M}_t)$ and let $\eta_M < \frac{1}{L_M}$:

$$\langle \nabla_{\boldsymbol{M}} f_{\mathcal{D}}(\boldsymbol{\alpha}_{t+1}, \boldsymbol{M}_t), \boldsymbol{M}_{t+1} - \boldsymbol{M}_t \rangle + \frac{L_M}{2} \left\| \boldsymbol{M}_{t+1} - \boldsymbol{M}_t \right\|_2^2$$

$$= -\eta_M \langle \nabla_{\boldsymbol{M}} f_{\mathcal{D}}(\boldsymbol{\alpha}_{t+1}, \boldsymbol{M}_t), \tilde{g}_{\mathcal{B}}(\boldsymbol{\alpha}_{t+1}, \boldsymbol{M}_t) \rangle + \frac{L_M \eta_M^2}{2} \left\| \tilde{g}_{\mathcal{B}}(\boldsymbol{\alpha}_{t+1}, \boldsymbol{M}_t) \right\|_2^2$$

$$= -\eta_M \langle \nabla_{\boldsymbol{M}} f_{\mathcal{D}}(\boldsymbol{\alpha}_{t+1}, \boldsymbol{M}_t), g_{\mathcal{D}}(\boldsymbol{\alpha}_{t+1}, \boldsymbol{M}_t) \rangle + \frac{L_M \eta_M^2}{2} \left\| \tilde{g}_{\mathcal{B}}(\boldsymbol{\alpha}_{t+1}, \boldsymbol{M}_t) \right\|_2^2$$

$$+ \eta_M \langle \nabla_{\boldsymbol{M}} f_{\mathcal{D}}(\boldsymbol{\alpha}_{t+1}, \boldsymbol{M}_t), g_{\mathcal{D}}(\boldsymbol{\alpha}_{t+1}, \boldsymbol{M}_t) - \tilde{g}_{\mathcal{B}}(\boldsymbol{\alpha}_{t+1}, \boldsymbol{M}_t) \rangle$$

$$\overset{(1)}{\leq} -\eta_M \left\| g_{\mathcal{D}}(\boldsymbol{\alpha}_{t+1}, \boldsymbol{M}_t) \right\|_2^2 + r(\boldsymbol{M}_t) - r(\boldsymbol{M}_{t+1}) + \frac{L_M \eta_M^2}{2} \left\| \tilde{g}_{\mathcal{B}}(\boldsymbol{\alpha}_{t+1}, \boldsymbol{M}_t) \right\|_2^2$$

$$+ \eta_M \left\langle \nabla_{\boldsymbol{M}} f_{\mathcal{D}}(\boldsymbol{\alpha}_{t+1}, \boldsymbol{M}_t) - \tilde{\nabla}_{\boldsymbol{M}} f_{\mathcal{B}}(\boldsymbol{\alpha}_{t+1}, \boldsymbol{M}_t), g_{\mathcal{D}}(\boldsymbol{\alpha}_{t+1}, \boldsymbol{M}_t) - \tilde{g}_{\mathcal{B}}(\boldsymbol{\alpha}_{t+1}, \boldsymbol{M}_t) \right\rangle$$

$$+ \eta_M \left\langle \tilde{\nabla}_{\boldsymbol{M}} f_{\mathcal{B}}(\boldsymbol{\alpha}_{t+1}, \boldsymbol{M}_t) - \tilde{g}_{\mathcal{B}}(\boldsymbol{\alpha}_{t+1}, \boldsymbol{M}_t), g_{\mathcal{D}}(\boldsymbol{\alpha}_{t+1}, \boldsymbol{M}_t) - \tilde{g}_{\mathcal{B}}(\boldsymbol{\alpha}_{t+1}, \boldsymbol{M}_t) \right\rangle$$

$$+ \eta_M \langle \tilde{g}_{\mathcal{B}}(\boldsymbol{\alpha}_{t+1}, \boldsymbol{M}_t), g_{\mathcal{D}}(\boldsymbol{\alpha}_{t+1}, \boldsymbol{M}_t) - \tilde{g}_{\mathcal{B}}(\boldsymbol{\alpha}_{t+1}, \boldsymbol{M}_t) \rangle$$

$$\overset{(2)}{\leq} -\eta_M \left\| g_{\mathcal{D}}(\boldsymbol{\alpha}_{t+1}, \boldsymbol{M}_t) \right\|_2^2 + r(\boldsymbol{M}_t) - r(\boldsymbol{M}_{t+1}) + \frac{L_M \eta_M^2}{2} \left\| \tilde{g}_{\mathcal{B}}(\boldsymbol{\alpha}_{t+1}, \boldsymbol{M}_t) \right\|_2^2$$

$$+ \eta_M \left\| \nabla_{\boldsymbol{M}} f_{\mathcal{D}}(\boldsymbol{\alpha}_{t+1}, \boldsymbol{M}_t) - \tilde{\nabla}_{\boldsymbol{M}} f_{\mathcal{B}}(\boldsymbol{\alpha}_{t+1}, \boldsymbol{M}_t) \right\|_2^2$$

$$+ \eta_M \left\langle \tilde{\nabla}_{\boldsymbol{M}} f_{\mathcal{B}}(\boldsymbol{\alpha}_{t+1}, \boldsymbol{M}_t) - \tilde{g}_{\mathcal{B}}(\boldsymbol{\alpha}_{t+1}, \boldsymbol{M}_t), g_{\mathcal{D}}(\boldsymbol{\alpha}_{t+1}, \boldsymbol{M}_t) - \tilde{g}_{\mathcal{B}}(\boldsymbol{\alpha}_{t+1}, \boldsymbol{M}_t) \right\rangle$$

$$+ \eta_M \langle \tilde{g}_{\mathcal{B}}(\boldsymbol{\alpha}_{t+1}, \boldsymbol{M}_t), g_{\mathcal{D}}(\boldsymbol{\alpha}_{t+1}, \boldsymbol{M}_t) - \tilde{g}_{\mathcal{B}}(\boldsymbol{\alpha}_{t+1}, \boldsymbol{M}_t) \rangle$$

$$\overset{(3)}{\leq} -\eta_M \left\| g_{\mathcal{D}}(\boldsymbol{\alpha}_{t+1}, \boldsymbol{M}_t) \right\|_2^2 + r(\boldsymbol{M}_t) - r(\boldsymbol{M}_{t+1}) + \frac{L_M \eta_M^2}{2} \left\| \tilde{g}_{\mathcal{B}}(\boldsymbol{\alpha}_{t+1}, \boldsymbol{M}_t) \right\|_2^2$$

$$+ \eta_M \left\| \nabla_{\boldsymbol{M}} f_{\mathcal{D}}(\boldsymbol{\alpha}_{t+1}, \boldsymbol{M}_t) - \tilde{\nabla}_{\boldsymbol{M}} f_{\mathcal{B}}(\boldsymbol{\alpha}_{t+1}, \boldsymbol{M}_t) \right\|_2^2$$

$$+ \eta_M \langle \tilde{g}_{\mathcal{B}} (\boldsymbol{\alpha}_{t+1}, \boldsymbol{M}_t), g_{\mathcal{D}} (\boldsymbol{\alpha}_{t+1}, \boldsymbol{M}_t) - \tilde{g}_{\mathcal{B}} (\boldsymbol{\alpha}_{t+1}, \boldsymbol{M}_t) \rangle$$

$$\overset{(4)}{=} -\eta_M \|g_{\mathcal{D}} (\boldsymbol{\alpha}_{t+1}, \boldsymbol{M}_t)\|_2^2 + r(\boldsymbol{M}_t) - r(\boldsymbol{M}_{t+1}) + \frac{L_M \eta_M^2}{2} \|\tilde{g}_{\mathcal{B}} (\boldsymbol{\alpha}_{t+1}, \boldsymbol{M}_t)\|_2^2$$

$$+ \eta_M \left\| \nabla_M f_{\mathcal{D}}(\boldsymbol{\alpha}_{t+1}, \boldsymbol{M}_t) - \tilde{\nabla}_M f_{\mathcal{B}}(\boldsymbol{\alpha}_{t+1}, \boldsymbol{M}_t) \right\|_2^2 + \frac{\eta_M}{2} \|g_{\mathcal{D}} (\boldsymbol{\alpha}_{t+1}, \boldsymbol{M}_t)\|_2^2$$

$$- \frac{\eta_M}{2} \|\tilde{g}_{\mathcal{B}} (\boldsymbol{\alpha}_{t+1}, \boldsymbol{M}_t)\|_2^2 - \frac{\eta_M}{2} \|g_{\mathcal{D}} (\boldsymbol{\alpha}_{t+1}, \boldsymbol{M}_t) - \tilde{g}_{\mathcal{B}} (\boldsymbol{\alpha}_{t+1}, \boldsymbol{M}_t)\|_2^2$$

$$= -\frac{\eta_M}{2} \|g_{\mathcal{D}} (\boldsymbol{\alpha}_{t+1}, \boldsymbol{M}_t)\|_2^2 + r(\boldsymbol{M}_t) - r(\boldsymbol{M}_{t+1}) + \left( \frac{L_M \eta_M^2}{2} - \frac{\eta_M}{2} \right) \|\tilde{g}_{\mathcal{B}} (\boldsymbol{\alpha}_{t+1}, \boldsymbol{M}_t)\|_2^2$$

$$+ \eta_M \left\| \nabla_M f_{\mathcal{D}}(\boldsymbol{\alpha}_{t+1}, \boldsymbol{M}_t) - \tilde{\nabla}_M f_{\mathcal{B}}(\boldsymbol{\alpha}_{t+1}, \boldsymbol{M}_t) \right\|_2^2 - \frac{\eta_M}{2} \|g_{\mathcal{D}} (\boldsymbol{\alpha}_{t+1}, \boldsymbol{M}_t) - \tilde{g}_{\mathcal{B}} (\boldsymbol{\alpha}_{t+1}, \boldsymbol{M}_t)\|_2^2$$

$$\overset{(5)}{\leq} -\frac{\eta_M}{2} \|g_{\mathcal{D}} (\boldsymbol{\alpha}_{t+1}, \boldsymbol{M}_t)\|_2^2 + r(\boldsymbol{M}_t) - r(\boldsymbol{M}_{t+1})$$

$$+ \left( \frac{L_M \eta_M^2}{4} - \frac{\eta_M}{4} \right) \left( \|g_{\mathcal{D}} (\boldsymbol{\alpha}_{t+1}, \boldsymbol{M}_t)\|_2^2 - 2 \|g_{\mathcal{D}} (\boldsymbol{\alpha}_{t+1}, \boldsymbol{M}_t) - \tilde{g}_{\mathcal{B}} (\boldsymbol{\alpha}_{t+1}, \boldsymbol{M}_t)\|_2^2 \right)$$

$$+ \eta_M \left\| \nabla_M f_{\mathcal{D}}(\boldsymbol{\alpha}_{t+1}, \boldsymbol{M}_t) - \tilde{\nabla}_M f_{\mathcal{B}}(\boldsymbol{\alpha}_{t+1}, \boldsymbol{M}_t) \right\|_2^2 - \frac{\eta_M}{2} \cdot \|g_{\mathcal{D}} (\boldsymbol{\alpha}_{t+1}, \boldsymbol{M}_t) - \tilde{g}_{\mathcal{B}} (\boldsymbol{\alpha}_{t+1}, \boldsymbol{M}_t)\|_2^2$$

$$= \left( \frac{L_M \eta_M^2}{4} - \frac{3\eta_M}{4} \right) \|g_{\mathcal{D}} (\boldsymbol{\alpha}_{t+1}, \boldsymbol{M}_t)\|_2^2 + r(\boldsymbol{M}_t) - r(\boldsymbol{M}_{t+1})$$

$$- \frac{L_M \eta_M^2}{2} \|g_{\mathcal{D}} (\boldsymbol{\alpha}_{t+1}, \boldsymbol{M}_t) - \tilde{g}_{\mathcal{B}} (\boldsymbol{\alpha}_{t+1}, \boldsymbol{M}_t)\|_2^2 + \eta_M \left\| \nabla_M f_{\mathcal{D}}(\boldsymbol{\alpha}_{t+1}, \boldsymbol{M}_t) - \tilde{\nabla}_M f_{\mathcal{B}}(\boldsymbol{\alpha}_{t+1}, \boldsymbol{M}_t) \right\|_2^2$$

$$\overset{(6)}{\leq} -\frac{\eta_M}{2} \|g_{\mathcal{D}} (\boldsymbol{\alpha}_{t+1}, \boldsymbol{M}_t)\|_2^2 + r(\boldsymbol{M}_t) - r(\boldsymbol{M}_{t+1}) + \eta_M \left\| \nabla_M f_{\mathcal{D}}(\boldsymbol{\alpha}_{t+1}, \boldsymbol{M}_t) - \tilde{\nabla}_M f_{\mathcal{B}}(\boldsymbol{\alpha}_{t+1}, \boldsymbol{M}_t) \right\|_2^2$$

$$\overset{(7)}{\leq} -\frac{\eta_M}{2} \|g_{\mathcal{D}} (\boldsymbol{\alpha}_{t+1}, \boldsymbol{M}_t)\|_2^2 + r(\boldsymbol{M}_t) - r(\boldsymbol{M}_{t+1}) + 2\eta_M \left\| \nabla_M f_{\mathcal{D}}(\boldsymbol{\alpha}_{t+1}, \boldsymbol{M}_t) - \tilde{\nabla}_M f_{\mathcal{D}}(\boldsymbol{\alpha}_{t+1}, \boldsymbol{M}_t) \right\|_2^2$$

$$+ 2\eta_M \left\| \tilde{\nabla}_M f_{\mathcal{D}}(\boldsymbol{\alpha}_{t+1}, \boldsymbol{M}_t) - \tilde{\nabla}_M f_{\mathcal{B}}(\boldsymbol{\alpha}_{t+1}, \boldsymbol{M}_t) \right\|_2^2. \tag{46}$$

**Note:**

- (1) use **Lemma 1** in (Ghadimi & Lan, 2013).

- (2) is because:

$$\left\langle \nabla_M f_{\mathcal{D}}(\boldsymbol{\alpha}_{t+1}, \boldsymbol{M}_t) - \tilde{\nabla}_M f_{\mathcal{B}}(\boldsymbol{\alpha}_{t+1}, \boldsymbol{M}_t), g_{\mathcal{D}} (\boldsymbol{\alpha}_{t+1}, \boldsymbol{M}_t) - \tilde{g}_{\mathcal{B}} (\boldsymbol{\alpha}_{t+1}, \boldsymbol{M}_t) \right\rangle$$

$$\leq \left\| \left\langle \nabla_M f_{\mathcal{D}}(\boldsymbol{\alpha}_{t+1}, \boldsymbol{M}_t) - \tilde{\nabla}_M f_{\mathcal{B}}(\boldsymbol{\alpha}_{t+1}, \boldsymbol{M}_t), g_{\mathcal{D}} (\boldsymbol{\alpha}_{t+1}, \boldsymbol{M}_t) - \tilde{g}_{\mathcal{B}} (\boldsymbol{\alpha}_{t+1}, \boldsymbol{M}_t) \right\rangle \right\|_2$$

$$\leq \left\| \nabla_M f_{\mathcal{D}}(\boldsymbol{\alpha}_{t+1}, \boldsymbol{M}_t) - \tilde{\nabla}_M f_{\mathcal{B}}(\boldsymbol{\alpha}_{t+1}, \boldsymbol{M}_t) \right\|_2 \cdot \|g_{\mathcal{D}} (\boldsymbol{\alpha}_{t+1}, \boldsymbol{M}_t) - \tilde{g}_{\mathcal{B}} (\boldsymbol{\alpha}_{t+1}, \boldsymbol{M}_t)\|_2$$

$$\leq \left\| \nabla_M f_{\mathcal{D}}(\boldsymbol{\alpha}_{t+1}, \boldsymbol{M}_t) - \tilde{\nabla}_M f_{\mathcal{B}}(\boldsymbol{\alpha}_{t+1}, \boldsymbol{M}_t) \right\|_2^2. \tag{47}$$

The second inequality use: $\|ab\|_2 \leq \|a\|_2 \cdot \|b\|_2$; The third inequality uses **Proposition 1** in (Ghadimi & Lan, 2013).

- (3) is because:

$$\left\langle \tilde{\nabla}_M f_{\mathcal{B}}(\boldsymbol{\alpha}_{t+1}, \boldsymbol{M}_t) - \tilde{g}_{\mathcal{B}} (\boldsymbol{\alpha}_{t+1}, \boldsymbol{M}_t), g_{\mathcal{D}} (\boldsymbol{\alpha}_{t+1}, \boldsymbol{M}_t) - \tilde{g}_{\mathcal{B}} (\boldsymbol{\alpha}_{t+1}, \boldsymbol{M}_t) \right\rangle$$

$$= \frac{1}{\eta_M^2} \left\langle \eta_M \tilde{\nabla}_M f_{\mathcal{B}}(\boldsymbol{\alpha}_{t+1}, \boldsymbol{M}_t) - \eta_M \tilde{g}_{\mathcal{B}} (\boldsymbol{\alpha}_{t+1}, \boldsymbol{M}_t), \eta_M g_{\mathcal{D}} (\boldsymbol{\alpha}_{t+1}, \eta_M \boldsymbol{M}_t) - \eta_M \tilde{g}_{\mathcal{B}} (\boldsymbol{\alpha}_{t+1}, \boldsymbol{M}_t) \right\rangle$$

$$= \frac{1}{\eta_M^2} \left\langle \boldsymbol{M}_{t+1} - \left[ \boldsymbol{M}_t - \eta_M \tilde{\nabla}_M f_{\mathcal{B}}(\boldsymbol{\alpha}_{t+1}, \boldsymbol{M}_t) \right], \boldsymbol{M}_{t+1} - [\boldsymbol{M}_t - \eta_M g_{\mathcal{D}} (\boldsymbol{\alpha}_{t+1}, \eta_M \boldsymbol{M}_t)] \right\rangle$$

$$\leq 0. \tag{48}$$

The second equality in (48) use the definitions (22), (24) and (25); the inequality in (48) use Bourbaki-Cheney-Goldstein inequality (Holmes, 1973, Eq. (1.5)) and the definitions:

$$\boldsymbol{M}_{t+1} = \text{prox}_{\eta_M r}\left[\boldsymbol{M}_t - \eta_M \tilde{\nabla}_M f_{\mathcal{B}}(\boldsymbol{\alpha}_{t+1}, \boldsymbol{M}_t)\right],$$

$$\boldsymbol{M}_t - \eta_M g_{\mathcal{D}}(\boldsymbol{\alpha}_{t+1}, \eta_M \boldsymbol{M}_t) = \text{prox}_{\eta_M r}\left[\boldsymbol{M}_t - \eta_M \cdot \nabla_M f_{\mathcal{D}}(\boldsymbol{\alpha}_{t+1}, \boldsymbol{M}_t)\right].$$

- (4) use equality: $ab = \frac{(a+b)^2 - a^2 - b^2}{2}$.

- (5) use inequality: $a^2 \geq \frac{(a+b)^2 - 2b^2}{2}$ and $\eta_M < \frac{1}{L_M}$.

- (6) use $\eta_M < \frac{1}{L_M}$.

- (7) use inequality: $\|a + b\|_2^2 \leq 2\|a\|_2^2 + 2\|b\|_2^2$.

Then we take expectations $\mathbb{E}_{\mathcal{B}}$ and $\mathbb{E}_{\{\Phi^s \sim \mathcal{S}(\boldsymbol{\alpha})\}_{s=1}^{I_M}}$ on both sides of (46), and abbreviate $\mathbb{E}_{\{\Phi^s \sim \mathcal{S}(\boldsymbol{\alpha})\}_{s=1}^{I_M}}$ as $\mathbb{E}_{\{\Phi^s\}_{s=1}^{I_M}}$:

$$\mathbb{E}_{\mathcal{B}}\mathbb{E}_{\{\Phi^s\}_{s=1}^{I_M}}\left[\langle \nabla_M f_{\mathcal{D}}(\boldsymbol{\alpha}_{t+1}, \boldsymbol{M}_t), \boldsymbol{M}_{t+1} - \boldsymbol{M}_t\rangle + \frac{L_M}{2}\|\boldsymbol{M}_{t+1} - \boldsymbol{M}_t\|_2^2\right]$$

$$\leq -\frac{\eta_M}{2}\|g_{\mathcal{D}}(\boldsymbol{\alpha}_{t+1}, \boldsymbol{M}_t)\|_2^2 + r(\boldsymbol{M}_t) - r(\boldsymbol{M}_{t+1})$$

$$+ 2\eta_M \mathbb{E}_{\mathcal{B}}\mathbb{E}_{\{\Phi^s\}_{s=1}^{I_M}}\left\|\nabla_M f_{\mathcal{D}}(\boldsymbol{\alpha}_{t+1}, \boldsymbol{M}_t) - \tilde{\nabla}_M f_{\mathcal{D}}(\boldsymbol{\alpha}_{t+1}, \boldsymbol{M}_t)\right\|_2^2$$

$$+ 2\eta_M \mathbb{E}_{\{\Phi^s\}_{s=1}^{I_M}}\mathbb{E}_{\mathcal{B}}\left\|\tilde{\nabla}_M f_{\mathcal{D}}(\boldsymbol{\alpha}_{t+1}, \boldsymbol{M}_t) - \tilde{\nabla}_M f_{\mathcal{B}}(\boldsymbol{\alpha}_{t+1}, \boldsymbol{M}_t)\right\|_2^2$$

$$\overset{(1)}{\leq} -\frac{\eta_M}{2}\|g_{\mathcal{D}}(\boldsymbol{\alpha}_{t+1}, \boldsymbol{M}_t)\|_2^2 + r(\boldsymbol{M}_t) - r(\boldsymbol{M}_{t+1}) + 2\eta_M \mathbb{E}_{\mathcal{B}}\left[\frac{\tilde{\sigma}_M^2}{I_M}\right] + 2\eta_M \mathbb{E}_{\{\Phi^s\}_{s=1}^{I_M}}\left[\frac{\sigma_M^2}{B}\right]$$

$$= -\frac{\eta_M}{2}\|g_{\mathcal{D}}(\boldsymbol{\alpha}_{t+1}, \boldsymbol{M}_t)\|_2^2 + r(\boldsymbol{M}_t) - r(\boldsymbol{M}_{t+1}) + \frac{2\eta_M \tilde{\sigma}_M^2}{I_M} + \frac{2\eta_M \sigma_M^2}{B}. \tag{49}$$

where (1) use use the bounded variance of stochastic gradients and gradient with prompt sampling for $\boldsymbol{M}$ in **Assumption 1** and **Lemma 4**.

We take expectations $\mathbb{E}_{\mathcal{B}}$, $\mathbb{E}_{\{\Phi^r\}_{r=1}^{I_\alpha}}$ and $\mathbb{E}_{\{\Phi^s\}_{s=1}^{I_M}}$ for (44), then substitute (45), (49) into (44) and both sides accumulate with respect to $t = 0, 1, \cdots, T-1$ and divide by $T$:

$$\frac{1}{T}\sum_{t=0}^{T-1}\mathbb{E}_{\mathcal{B}}\mathbb{E}_{\{\Phi^r\}_{r=1}^{I_\alpha}}\mathbb{E}_{\{\Phi^s\}_{s=1}^{I_M}}\left[f_{\mathcal{D}}(\boldsymbol{\alpha}_{t+1}, \boldsymbol{M}_{t+1}) - f_{\mathcal{D}}(\boldsymbol{\alpha}_t, \boldsymbol{M}_t)\right]$$

$$\leq \frac{1}{T}\sum_{t=0}^{T-1}\sum_{i=1}^{n}\mathbb{E}_{\mathcal{B}}\mathbb{E}_{\{\Phi^r\}_{r=1}^{I_\alpha}}\left[\langle \nabla_{\boldsymbol{\alpha}_i} f_{\mathcal{D}}(\boldsymbol{\alpha}_t, \boldsymbol{M}_t), \boldsymbol{\alpha}_{i,t+1} - \boldsymbol{\alpha}_{i,t}\rangle + \frac{L_\alpha}{2}\|\boldsymbol{\alpha}_{i,t+1} - \boldsymbol{\alpha}_{i,t}\|_2^2\right]$$

$$+ \frac{1}{T}\sum_{t=0}^{T-1}\mathbb{E}_{\mathcal{B}}\mathbb{E}_{\{\Phi^s\}_{s=1}^{I_M}}\langle \nabla_M f_{\mathcal{D}}(\boldsymbol{\alpha}_{t+1}, \boldsymbol{M}_t), \boldsymbol{M}_{t+1} - \boldsymbol{M}_t\rangle + \frac{L_M}{2}\|\boldsymbol{M}_{t+1} - \boldsymbol{M}_t\|_2^2$$

$$\leq \frac{1}{T}\sum_{t=0}^{T-1}\sum_{i=1}^{n}\left[-\frac{\eta_\alpha}{2}\|\nabla_{\boldsymbol{\alpha}_i} f_{\mathcal{D}}(\boldsymbol{\alpha}_t, \boldsymbol{M}_t)\|_2^2 + \frac{L_\alpha \eta_\alpha^2 \tilde{\sigma}_\alpha^2}{I_\alpha^2} + \frac{L_\alpha \eta_\alpha^2 \sigma_\alpha^2}{B}\right]$$

$$\frac{1}{T}\sum_{t=0}^{T-1}\left[-\frac{\eta_M}{2}\|g_{\mathcal{D}}(\boldsymbol{\alpha}_{t+1}, \boldsymbol{M}_t)\|_2^2 + r(\boldsymbol{M}_t) - r(\boldsymbol{M}_{t+1}) + \frac{2\eta_M \tilde{\sigma}_M^2}{I_M} + \frac{2\eta_M \sigma_M^2}{B}\right]$$

$$\overset{(1)}{=} -\frac{\eta_\alpha}{2}\frac{1}{T}\sum_{t=0}^{T-1}\|\nabla_{\boldsymbol{\alpha}} f_{\mathcal{D}}(\boldsymbol{\alpha}_t, \boldsymbol{M}_t)\|_2^2 + \frac{nL_\alpha \eta_\alpha^2 \tilde{\sigma}_\alpha^2}{I_\alpha^2} + \frac{nL_\alpha \eta_\alpha^2 \sigma_\alpha^2}{B}$$

$$- \frac{\eta_M}{2}\frac{1}{T}\sum_{t=0}^{T-1}\|g_{\mathcal{D}}(\boldsymbol{\alpha}_{t+1}, \boldsymbol{M}_t)\|_2^2 + \frac{1}{T}\sum_{t=0}^{T-1}\left[r(\boldsymbol{M}_t) - r(\boldsymbol{M}_{t+1})\right] + \frac{2\eta_M \tilde{\sigma}_M^2}{I_M} + \frac{2\eta_M \sigma_M^2}{B}. \tag{50}$$

where (1) is because $\boldsymbol{\alpha} = (\boldsymbol{\alpha}_1, \cdots, \boldsymbol{\alpha}_i, \cdots \boldsymbol{\alpha}_n)$.

Then we organize the inequality (50):

$$
\frac{\eta_{\boldsymbol{\alpha}}}{2} \frac{1}{T} \sum_{t=0}^{T-1} \|\nabla_{\boldsymbol{\alpha}} f_{\mathcal{D}}(\boldsymbol{\alpha}_t, \boldsymbol{M}_t)\|_2^2 + \frac{\eta_{\boldsymbol{M}}}{2} \frac{1}{T} \sum_{t=0}^{T-1} \|g_{\mathcal{D}}(\boldsymbol{\alpha}_{t+1}, \boldsymbol{M}_t)\|_2^2
$$

$$
\leq \frac{1}{T} \sum_{t=0}^{T-1} [F(\boldsymbol{\alpha}_t, \boldsymbol{M}_t; \boldsymbol{X}) - F(\boldsymbol{\alpha}_{t+1}, \boldsymbol{M}_{t+1}; \boldsymbol{X})] + \frac{nL_{\boldsymbol{\alpha}} \eta_{\boldsymbol{\alpha}}^2 \tilde{\sigma}_{\boldsymbol{\alpha}}^2}{I_{\boldsymbol{\alpha}}^2} + \frac{2\eta_{\boldsymbol{M}} \tilde{\sigma}_{\boldsymbol{M}}^2}{I_{\boldsymbol{M}}} + \frac{nL_{\boldsymbol{\alpha}} \eta_{\boldsymbol{\alpha}}^2 \sigma_{\boldsymbol{\alpha}}^2 + 2\eta_{\boldsymbol{M}} \sigma_{\boldsymbol{M}}^2}{B}
$$

$$
= \frac{F(\boldsymbol{\alpha}_0, \boldsymbol{M}_0; \boldsymbol{X}) - F(\boldsymbol{\alpha}_T, \boldsymbol{M}_T; \boldsymbol{X})}{T} + \frac{nL_{\boldsymbol{\alpha}} \eta_{\boldsymbol{\alpha}}^2 \tilde{\sigma}_{\boldsymbol{\alpha}}^2}{I_{\boldsymbol{\alpha}}^2} + \frac{2\eta_{\boldsymbol{M}} \tilde{\sigma}_{\boldsymbol{M}}^2}{I_{\boldsymbol{M}}} + \frac{nL_{\boldsymbol{\alpha}} \eta_{\boldsymbol{\alpha}}^2 \sigma_{\boldsymbol{\alpha}}^2 + 2\eta_{\boldsymbol{M}} \sigma_{\boldsymbol{M}}^2}{B}
$$

$$
\overset{(1)}{\leq} \frac{F(\boldsymbol{\alpha}_0, \boldsymbol{M}_0; \boldsymbol{X}) - F(\boldsymbol{\alpha}_*, \boldsymbol{M}_*; \boldsymbol{X})}{T} + \frac{nL_{\boldsymbol{\alpha}} \eta_{\boldsymbol{\alpha}}^2 \tilde{\sigma}_{\boldsymbol{\alpha}}^2}{I_{\boldsymbol{\alpha}}^2} + \frac{2\eta_{\boldsymbol{M}} \tilde{\sigma}_{\boldsymbol{M}}^2}{I_{\boldsymbol{M}}} + \frac{nL_{\boldsymbol{\alpha}} \eta_{\boldsymbol{\alpha}}^2 \sigma_{\boldsymbol{\alpha}}^2 + 2\eta_{\boldsymbol{M}} \sigma_{\boldsymbol{M}}^2}{B}
$$

$$
\overset{(2)}{\leq} \frac{\triangle}{T} + \frac{nL_{\boldsymbol{\alpha}} \eta_{\boldsymbol{\alpha}}^2 \tilde{\sigma}_{\boldsymbol{\alpha}}^2}{I_{\boldsymbol{\alpha}}^2} + \frac{2\eta_{\boldsymbol{M}} \tilde{\sigma}_{\boldsymbol{M}}^2}{I_{\boldsymbol{M}}} + \frac{nL_{\boldsymbol{\alpha}} \eta_{\boldsymbol{\alpha}}^2 \sigma_{\boldsymbol{\alpha}}^2 + 2\eta_{\boldsymbol{M}} \sigma_{\boldsymbol{M}}^2}{B}.
$$

where (1) because the objective function is non-convex, thus $F(\boldsymbol{\alpha}_*, \boldsymbol{M}_*; \boldsymbol{X}) \leq F(\boldsymbol{\alpha}_T, \boldsymbol{M}_T; \boldsymbol{X})$;

(2) use **Assumption 2**.

We let $\eta_{min} = \min\{\eta_{\boldsymbol{\alpha}}, \eta_{\boldsymbol{M}}\}$, $\eta_{max} = \max\{\eta_{\boldsymbol{\alpha}}, \eta_{\boldsymbol{M}}\}$ and $q_\eta = \frac{\eta_{max}}{\eta_{min}} < \infty$:

$$
\frac{1}{T} \sum_{t=0}^{T-1} \left( \|\nabla_{\boldsymbol{\alpha}} f_{\mathcal{D}}(\boldsymbol{\alpha}_t, \boldsymbol{M}_t)\|_2^2 + \|g_{\mathcal{D}}(\boldsymbol{\alpha}_{t+1}, \boldsymbol{M}_t)\|_2^2 \right)
$$

$$
\leq \frac{2\triangle}{T\eta_{min}} + \frac{2nL_{\boldsymbol{\alpha}} \eta_{\boldsymbol{\alpha}} q_\eta \tilde{\sigma}_{\boldsymbol{\alpha}}^2}{I_{\boldsymbol{\alpha}}^2} + \frac{4q_\eta \tilde{\sigma}_{\boldsymbol{M}}^2}{I_{\boldsymbol{M}}} + \frac{2nL_{\boldsymbol{\alpha}} \eta_{\boldsymbol{\alpha}} q_\eta \sigma_{\boldsymbol{\alpha}}^2 + 4q_\eta \sigma_{\boldsymbol{M}}^2}{B}
$$

$$
\overset{(1)}{\leq} \frac{2\triangle}{T\eta_{min}} + \frac{2nq_\eta \tilde{\sigma}_{\boldsymbol{\alpha}}^2}{I_{\boldsymbol{\alpha}}^2} + \frac{4q_\eta \tilde{\sigma}_{\boldsymbol{M}}^2}{I_{\boldsymbol{M}}} + \frac{2nq_\eta \sigma_{\boldsymbol{\alpha}}^2 + 4q_\eta \sigma_{\boldsymbol{M}}^2}{B}.
$$

where (1) use $\eta_{\boldsymbol{\alpha}} < \frac{1}{L_{\boldsymbol{\alpha}}}$. $\qquad \square$

**Corollary 1** (Convergence complexity of LEAP). *Suppose **Assumption 1**, **2** and **3** hold, and run **Algorithm 1** with $\eta_{\boldsymbol{\alpha}} = \frac{c_1}{L_{\boldsymbol{\alpha}}}(0 < c_1 < 1)$, $\eta_{\boldsymbol{M}} = \frac{c_2}{L_{\boldsymbol{M}}}(0 < c_1 < 1)$, $\eta_{min} = \min\left\{\frac{c_1}{L_{\boldsymbol{\alpha}}}, \frac{c_2}{L_{\boldsymbol{M}}}\right\}$, $q_\eta = \max\left\{\frac{c_1}{c_2}, \frac{c_2}{c_1}\right\} < \infty$, $B = \frac{8nq_\eta \sigma_{\boldsymbol{\alpha}}^2 + 16q_\eta \sigma_{\boldsymbol{M}}^2}{\epsilon^2}$, $I_{\boldsymbol{\alpha}} = \frac{\sqrt{8nq_\eta \tilde{\sigma}_{\boldsymbol{\alpha}}^2}}{\epsilon}$, $I_{\boldsymbol{M}} = \frac{16q_\eta \tilde{\sigma}_{\boldsymbol{M}}^2}{\epsilon^2}$ and $T = \frac{8\triangle}{\eta_{min} \epsilon^2}$, then the output of **Algorithm 1** satisfies:*

$$
\frac{1}{T} \sum_{t=0}^{T-1} \left( \|\nabla_{\boldsymbol{\alpha}} f_{\mathcal{D}}(\boldsymbol{\alpha}_t, \boldsymbol{M}_t)\|_2^2 + \|g_{\mathcal{D}}(\boldsymbol{\alpha}_{t+1}, \boldsymbol{M}_t)\|_2^2 \right) \leq \epsilon^2. \tag{51}
$$

*Thus, the total oracle complexity for LEAP is $\mathcal{O}\left(\frac{1}{\epsilon^4}\right)$.*

*Proof.* To ensure an $\epsilon$-solution:

$$
\frac{1}{T} \sum_{t=0}^{T-1} \left( \|\nabla_{\boldsymbol{\alpha}} f_{\mathcal{D}}(\boldsymbol{\alpha}_t, \boldsymbol{M}_t)\|_2^2 + \|g_{\mathcal{D}}(\boldsymbol{\alpha}_{t+1}, \boldsymbol{M}_t)\|_2^2 \right) \leq \epsilon^2.
$$

Then, we let:

$$
\frac{2\triangle}{T\eta_{min}} = \frac{\epsilon^2}{4}, \qquad \frac{2nq_\eta \tilde{\sigma}_{\boldsymbol{\alpha}}^2}{I_{\boldsymbol{\alpha}}^2} = \frac{\epsilon^2}{4}, \qquad \frac{4q_\eta \tilde{\sigma}_{\boldsymbol{M}}^2}{I_{\boldsymbol{M}}} = \frac{\epsilon^2}{4}, \qquad \frac{2nq_\eta \sigma_{\boldsymbol{\alpha}}^2 + 4q_\eta \sigma_{\boldsymbol{M}}^2}{B} = \frac{\epsilon^2}{4}.
$$

Finally, solving the above system of equations gives:

$$
B = \frac{8nq_\eta \sigma_{\boldsymbol{\alpha}}^2 + 16q_\eta \sigma_{\boldsymbol{M}}^2}{\epsilon^2}, \qquad I_{\boldsymbol{\alpha}} = \frac{\sqrt{8nq_\eta \tilde{\sigma}_{\boldsymbol{\alpha}}^2}}{\epsilon}, \qquad I_{\boldsymbol{M}} = \frac{16q_\eta \tilde{\sigma}_{\boldsymbol{M}}^2}{\epsilon^2}, \qquad T = \frac{8\triangle}{\eta_{min} \epsilon^2}.
$$

$\qquad \square$

## A.6 DATASETS

- **ELEC & BOOK:** The Amazon-Electronics (ELEC) and Amazon-Books (BOOK) dataset are collections of user reviews extracted from the electronics category on the Amazon platform, widely used in research on natural language processing and recommendation systems (McAuley et al., 2015). This dataset includes a large volume of user reviews on electronic products, encompassing review texts, ratings (typically ranging from 1 to 5 stars), product information, and users' purchase histories.

- **CoLA:** The Corpus of Linguistic Acceptability (CoLA), introduced by (Warstadt et al., 2019), consists of 8,500 training examples drawn from books and journal articles on linguistic theory. The task involves determining whether a given sentence is linguistically acceptable.

- **QNLI:** The Question Natural Language Inference (QNLI) task comprises 108,000 training examples derived from the Stanford Question Answering Dataset (SQuAD) (Rajpurkar et al., 2018). The objective of the task is to determine whether a given sentence contains the answer to a corresponding question.

- **RTE:** The Recognizing Textual Entailment (RTE) task (Dagan et al., 2005) includes 2,500 training examples sourced from various textual entailment challenges. The task involves determining whether a given premise sentence entails a corresponding hypothesis sentence.

- **SNLI:** The Stanford Natural Language Inference (SNLI) dataset is a widely used benchmark in the field of natural language processing, specifically designed for the task of Natural Language Inference (NLI). Created by (Bowman et al., 2015), the dataset comprises 570,000 manually annotated sentence pairs and aims to evaluate models' abilities to understand and reason about the logical relationships between sentences.

- **SST-2:** The Stanford Sentiment Treebank (SST) (Socher et al., 2013) contains 67,000 training examples of movie reviews with human-provided annotations. The task aims to determine whether a given sentence expresses a positive or negative sentiment.

- **AG:** The AG's news (AG) dataset is a widely used benchmark for text classification tasks in natural language processing. It consists of news articles collected from over 2,000 news sources, divided into four distinct categories: World, Sports, Business, and Science/Technology (Zhang et al., 2015). The dataset includes 120,000 training examples and 7,600 test examples, with each example being a short news article headline and description.

Table 5: Summary statistics of the experimental datasets. # Class, # Train, # Dev, and # Test denote the number of classes, training set, development set, and test set, respectively.

| Dataset | # Class | # Train | # Dev | # Test | Domain |
|---------|---------|---------|-------|--------|--------|
| BOOK | 2 | 55.6k | 7.9k | 26.0k | Amazon |
| CoLA | 2 | 8.6k | 1.0k | 1.0k | Books, Articles |
| ELEC | 2 | 10.8k | 1.5k | 3.1k | Amazon |
| QNLI | 2 | 104.7k | 5.5k | 5.5k | Wikipedia |
| RTE | 2 | 2.5k | 277 | 3.0k | News, Wikipedia |
| SNLI | 3 | 550.2k | 10k | 10k | Novels, Reports |
| SST-2 | 2 | 67.3k | 872 | 1821 | Movie Review |
| AG | 4 | 120.0k | - | 7.6k | News, Reports |

## A.7 EXPERIMENTAL DETAILS

**Hyperparameters.** The main hyperparameters of our algorithm is given in Table 6.

Table 6: Main hyperparameters used in our algorithm.

| Hyperparameter | RoBERTa-large | GPT2-XL | Llama3 |
|---|---|---|---|
| query limit | 4000 | 2000 | 1000 |
| train batch size | 32 | 16 | 8 |
| eval batch size | 32 | 4 | 4 |
| $\zeta$ | | 16 | |
| $N$ | | 100 | |
| $n$ | | $\{50, 20\}$ | |
| $I_\alpha$ | | 20 | |
| $\eta_\alpha$ | | 1e-2 | |
| $I_M$ | | 20 | |
| $\eta_M$ | | 1e-3 | |

**Manual Templates.** The templates used for our approach and baselines are given in Table 7.

Table 7: Input templates used in RoBERTa-large, GPT2-XL, and Llama3. $\langle$Sentence$\rangle$ denotes the sentences in the dataset. [MASK] denotes the mask token of RoBERTa-large.

| Dataset | RoBERTa-large | GPT2-XL / Llama3 |
|---|---|---|
| BOOK | $\langle$Sentence$_1\rangle$. It was [MASK]. | $\langle$Sentence$_1\rangle$. It was |
| CoLA | $\langle$Sentence$_1\rangle$. correct? [MASK]. | $\langle$Sentence$_1\rangle$. correct? |
| ELEC | $\langle$Sentence$_1\rangle$. It was [MASK]. | $\langle$Sentence$_1\rangle$. It was |
| QNLI | $\langle$Sentence$_1\rangle$ entailment? [MASK], $\langle$Sentence$_2\rangle$. | $\langle$Sentence$_1\rangle$ entailment? $\langle$Sentence$_2\rangle$. |
| RTE | $\langle$Sentence$_1\rangle$ entailment? [MASK], $\langle$Sentence$_2\rangle$. | $\langle$Sentence$_1\rangle$ entailment? $\langle$Sentence$_2\rangle$. |
| SNLI | $\langle$Sentence$_1\rangle$ entailment? [MASK], $\langle$Sentence$_2\rangle$. | $\langle$Sentence$_1\rangle$ entailment? $\langle$Sentence$_2\rangle$. |
| SST-2 | $\langle$Sentence$_1\rangle$. It was [MASK]. | $\langle$Sentence$_1\rangle$. It was |
| AG | $\langle$Sentence$_1\rangle$. It was [MASK]. | $\langle$Sentence$_1\rangle$. It was |

## A.8 ADDITIONAL EXPERIMENTAL RESULTS

**Example prompts.** Some learned prompts of our method on the RoBERTa-large model are provided in Table 8.

Table 8: Example prompts of our method on the RoBERTa-large model. × denotes the samples that are incorrectly predicted, while ✓ denotes those that are correctly predicted after applying the learned prompts.

| Dataset | Prompt+Sentence | Prediction |
|---|---|---|
| BOOK | I don't want to tell the story, and ruin the purpose of this book. This was an AWESOME book. | × |
|  | way been get if have on for well read the is life well as his in really because all read  I don't want to tell the story, and ruin the purpose of this book. This was an AWESOME book. | ✓ |
| CoLA | The more you would want, the less you would eat. correct? | × |
|  | The He was from out believe much It your Tom't Bill believe go as it like he Who go The more you would want, the less you would eat. correct? | ✓ |
| ELEC | I was very pleased with this product. It worked beautifully.Unfortunately it could not be used with newer PDA's. | × |
|  | little It all more you you me very get really much all've more from It headphones much than your I was very pleased with this product.  It worked beautifully.Unfortunately it could not be used with newer PDA's. | ✓ |
| QNLI | What happened to his lab? His lab was torn down in 1904, and its contents were sold two years later to satisfy a debt. | × |
|  | to Where under A into more which they year made being also called with had part has being population A  What happened to his lab?His lab was torn down in 1904, and its contents were sold two years later to satisfy a debt. | ✓ |
| RTE | Pibul Songgram was the pro-Japanese military dictator of Thailand during World War 2. | × |
|  | could " In some two be last found they they last has 2 being with includingThe from but out  Pibul Songgram was the pro-Japanese military dictator of Thailand during World War 2. | ✓ |
| SNLI | How many feature structures categories can label the first daughter? | × |
|  | shirt People as a her out as is Three his sits and ball that has jumping in walking black The  How many feature structures categories can label the first daughter? | ✓ |
| SST-2 | The turkey would've been a far better title. | × |
|  | only new been an enough a action more us enough and good movies by what he up to a own The turkey would've been a far better title. | ✓ |
| AG | it's hampered by a lifetime-channel kind of plot and a lead actress who is out of her depth. | × |
|  | only new been an enough a action more us enough and good movies by what he up to through own it's hampered by a lifetime-channel kind of plot and a lead actress who is out of her depth. | ✓ |

