# OpenReview forum: "Overcoming Missing Label Vocabulary in Black-Box Discrete Prompt Learning"
_ICLR.cc/2025/Conference — ICLR 2025 Conference Withdrawn Submission_

### Official Review · Reviewer_u8A6 · 2024-10-27

**Soundness:** 2
**Presentation:** 2
**Contribution:** 1
**Rating:** 3
**Confidence:** 3

**Summary:**

This paper studies black-box discrete prompt learning when there does not exist a clear map between the task labels and the language model vocabularies. It desgins a reinforcement learning algorithm (LEAP) to optimize the discrete prompts and the mapping between model vocabularies and the task labels. Experiments and theoretical analysis show the effectiveness of LEAP on text classification tasks compared to exsiting balck-box prompt tuning methods.

**Strengths:**

1. LMaSS is am important direction to explore in the era of LLMs.
2. The paper provides theoretical supports of convergence for the proposed algorithm.

**Weaknesses:**

1. The main weakness of this paper is the practical significance of the "label-free" setting. For large language models (LLMs), tasks are generally unified in a generative framework, where most labels can be expressed as text (or at least as phrases). For instance, in the Amazon Books rating task mentioned in lines 68-69, numerical ratings can be converted to strings, tokenized, and treated as labels. Therefore, further clarification of the practical implications of this "label-free" setting would be necessary.

2. (Following Point 1) The "label-free" setting appears to be independent of specific tuning methods. Are there other studies that explore this setting? If so, discussing them in the Related Work section would be needed.

3. A secondary weakness of the paper is the limited applicability of the proposed method to various tasks. Section 3.1 suggests that the approach relies on predefined categories, and the experiments focus solely on simple text classification tasks. This approach thus appears restricted to text classification, while most real-world applications involve language generation tasks without predefined "categories" (though ground-truth label tokens do exist).

**Questions:**

See Weakness.

---

### Official Review · Reviewer_LLdx · 2024-10-28

**Soundness:** 3
**Presentation:** 2
**Contribution:** 2
**Rating:** 3
**Confidence:** 2

**Summary:**

This paper proposes a black-box discrete prompt optimization method to deal with the scenario where no label vocabulary is available. The method includes optimizing the Gumbel-Softmax parameterization process and a mapping matrix. Experiments on multiple classification datasets and models with different scales prove the efficacy of the method.

**Strengths:**

- It is an interesting scenario where no pre-defined labels are available when optimizing prompts.
- The authors give a theoretical analysis of the convergence of the proposed optimization process.

**Weaknesses:**

- The paper focuses on classification tasks. In the era of LLM, prompt optimization towards generation is more desirable. I would like to see if the authors can adapt their methods to generation tasks (e.g., question-answering).
- In Section 5.1, it is important to emphasize the method for conducting the label-free setting. However, this aspect is not addressed in the paper.
- The improvements on some datasets are trivial (e.g., BOOK, CoLA, QNLI). Significance tests are recommended to validate the effect of the proposed method.

**Questions:**

- I am confused about the B_Y in Figure 1. I do not understand what it includes. Just input text or text and label? Can you explain the loss function in line 180? I am confused about what is the objective of this loss.
- The label-free scenario is an important application scenario in this work. However, I do not know how do you define "label-free". Do you use label information in the prompt optimization process?
- I will keep my confidence low because I do not understand the above-mentioned claim in the paper. If the authors explain them in detail, I will consider increasing the score. However, the presentation is not good and the paper should be reframed.

---

### Official Review · Reviewer_LiNf · 2024-11-04

**Soundness:** 2
**Presentation:** 2
**Contribution:** 1
**Rating:** 3
**Confidence:** 3

**Summary:**

The paper studied the setting of black-blox prompt learning without a label vocabulary. To solve the missing label vocabulary problem, the author proposed to learn a mapping from LLM tokens to discrete labels. The mapping and the discrete prompts are trained jointly via policy gradient descent. Experiments on GPT-XL, RoBERTA-large and Llama3-8B show that the newly proposed LEAP algorithm performs better than baselines like BDPL, SSPT.

**Strengths:**

The paper's idea of learning a mapping from token to label is straightforward. The author also proposed a training algorithm with variance-reduced policy gradient descent.

**Weaknesses:**

The experimental settings are not realistic. The author only conducted experiments on datasets with at most 4 classes (as shown in Table 5). If you have less than four label classes, it is often straightforward to assign textual description of the labels and there is no need to consider the missing-label-vocabulary problem. On the other hand, the author conducted experiments on weak LLMs like RoBERTa, GPT2-XL and Llama3-8B. The author needs to conduct experiments on stronger LLMs like OpenAI's GPT-4o, or at least Llama3-70B. Black-box prompt learning are usually targeted for stronger LLMs. It is not clear if LEAP will still work.

**Questions:**

N/A

---

### Official Review · Reviewer_itSQ · 2024-11-04

**Soundness:** 3
**Presentation:** 2
**Contribution:** 2
**Rating:** 5
**Confidence:** 3

**Summary:**

The paper presents LEAP, a method for black-box discrete prompt learning without relying on a predefined label vocabulary. It employs an alternating optimization framework to learn prompt tokens and a mapping matrix for LLM outputs. The paper provides convergence analysis and experimental results showing LEAP's effectiveness on label-vocabulary-free datasets.

**Strengths:**

1. The paper introduces LEAP, a method for black-box prompt learning that does not require a predefined label vocabulary, offering a new solution for optimizing LLMs in scenarios with limited access to model internals.

2. LEAP is presented with a clear structure, detailing an alternating optimization strategy and a learnable mapping matrix. The paper includes a thorough theoretical analysis on the convergence of the proposed method.

3. The research tackles a practical challenge in applying LLMs, enhancing their adaptability in real-world applications where internal model parameters and gradients are inaccessible.

**Weaknesses:**

1. The paper focuses on classification tasks and does not provide experiment results for generation tasks, which may limit the assessment of LEAP's versatility across different NLP applications.

2. The paper does not include comparisons with the latest model optimization methods like BBTv2[1] and GDFO[2], which could impact the perceived novelty and competitiveness of the LEAP method.

3. The paper lacks details on the initialization of the learnable matrix M and its optimization's influence on model performance.

**Questions:**

1. Could the authors elaborate on the initialization process of the learnable mapping matrix and its sensitivity to different initializations?

2. How does LEAP compare with existing methods in terms of computational resources required, especially when scaling up to larger models or datasets?

---

### Note · Authors · 2025-02-10

I have read and agree with the venue's withdrawal policy on behalf of myself and my co-authors.

---

### Meta-Review · Area_Chair_RH8f · 2024-12-18

**Metareview:**

This paper studies the problem of discrete prompt learning for a black box LLM where no label vocabulary is available. It employs an alternating optimization strategy to learn discrete prompt tokens and a learnable matrix for LLM outputs.

Some of the reviewers appreciate the authors tackling the interesting and challenging application scenario. The provided theoretical supported is valuable. But there are some concerns before this paper is ready for publication. The major one is the experimental setting is not practical. The paper only studies the classification problem, would be encouraged to see the results on generation tasks. Stronger comparison methods should be included. Based on the assessment, it is concluded that the paper could not be accepted in its current form and would require a major revision.

**Additional Comments On Reviewer Discussion:**

No rebuttal provided.

---

### Decision · Program_Chairs · 2025-01-22

Reject